# End-to-End 4D Heart Mesh Recovery Across Full-Stack and Sparse Cardiac MRI

**Yihong Chen**                                                          *yihong.chen@epfl.ch*
*School of Computer and Communication Science, EPFL*

**Jiancheng Yang**                                                      *jiancheng.yang@aalto.fi*
*ELLIS Institute Finland, Aalto University, Finland*

**Deniz Sayin Mercadier**                                   *deniz.mercadier@epfl.ch@epfl.ch*
*School of Computer and Communication Science, EPFL*

**Hieu Le**                                                                 *hle40@charlotte.edu*
*University of North Carolina at Charlotte*

**Juerg Schwitter**                                                   *jurg.schwitter@chuv.ch*
*Center for Interventional MRI, CHUV*

**Pascal Fua**                                                            *pascal.fua@epfl.ch*
*School of Computer and Communication Science, EPFL*

**Reviewed on OpenReview:** *https://openreview.net/forum?id=9k00kN5yk2*

## Abstract

Reconstructing cardiac motion from CMR sequences is critical for diagnosis, prognosis, and intervention. Existing methods rely on complete CMR stacks to infer full heart motion, limiting their applicability during intervention when only sparse observations are available. We present *TetHeart*, the first end-to-end framework for unified 4D heart mesh recovery from both offline full-stack and intra-procedural sparse-slice observations. Our method leverages deformable tetrahedra to capture shape and motion in a coherent space shared across cardiac structures. Before a procedure, it initializes detailed, patient-specific heart meshes from high-quality full stacks, which can then be updated using whatever slices can be obtained in real time, down to a single slice during the procedure. *TetHeart* incorporates several key innovations: (i) an attentive slice-adaptive 2D–3D feature assembly mechanism that integrates information from arbitrary numbers of slices at any position; (ii) a distillation strategy to ensure accurate reconstruction under extreme sparsity; and (iii) a weakly supervised motion learning scheme requiring annotations only at keyframes, such as the end-diastolic and end-systolic phases. Trained and validated on three large public datasets, and evaluated on additional private interventional and public datasets without retraining, *TetHeart* achieves state-of-the-art accuracy in both pre- and intra-procedural settings. Code and dataset is available at `https://github.com/Scalsol/TetHeart`.

## 1 Introduction

Modeling cardiac shape and motion is a vital 4D reconstruction problem with direct implications for medical imaging, motion analysis, and image-guided intervention Sanz & Fayad (2008); Li et al. (2024); Qiao et al. (2025); Qian et al. (2025). Cardiac magnetic resonance (CMR) provides high-quality temporal anatomy and has been widely used for both shape reconstruction Bai et al. (2015); Ye et al. (2023); Dou et al. (2024); Yang et al. (2024); Xiao et al. (2024) and motion estimation Qin et al. (2018); Meng et al. (2022a;b; 2023); Yuan et al. (2023). However, as shown in Fig. 1 (a), existing methods rely on full volumetric CMR stacks and

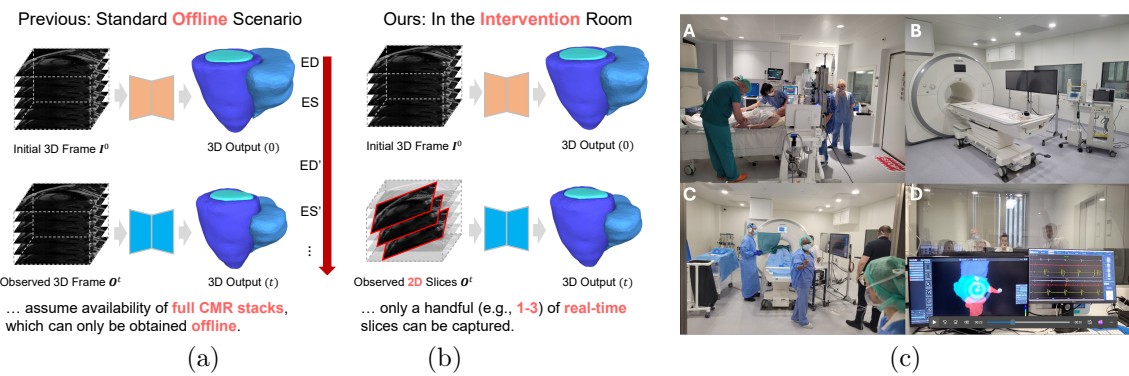

Figure 1: *Cardiac motion reconstruction.* (a) Standard offline scenario. (b) Intervention room setting. (c) Interventional MRI setup at our collaborating hospital: A. Preparation room located next to the iCMR room. B. iCMR room equipped with screens to visualize the heart and catheters during the intervention. C. Preparation of the intubated and ventilated patient inside the iCMR. D. Control room with visualizations.

are thus restricted to offline analysis and pre-operative planning. In interventional settings, as depicted by Fig. 1 (b), only a handful of low-resolution slices can be acquired in real time Gao et al. (2012); Nayak et al. (2022); Rogers et al. (2023), which is a limitation we observed in practice at our collaborating university hospital even though state-of-the-art MRI equipment is available, as shown in Fig. 1(c).

This limitation makes current 4D modeling pipelines unusable *during interventions.* Currently, interventional MRI technology to guide electrophysiological ablations is gaining increasing attention among electrophysiologists and this technique is used in more than ten countries. The most frequently performed iCMR-guided ablation nowadays is right atrial flutter ablation Paetsch et al. (2019) with an estimated number of over 400 cases performed so far. In addition, left ventricular arrhythmia ablations have also been performed recently Götte et al. (2025). Thus, making 3D modeling fast enough for use in such an interventional MRI setting is what motivates us to go beyond the current state of the art and to recover full 3D shape and motion from sparse and partial observations. Our goal is to enable real-time tracking of patient-specific heart dynamics, with a view to delivering guidance to interventional teams.

To this end, we introduce *TetHeart*, a unified framework for 4D cardiac motion reconstruction that operates consistently across both full-stack and sparse-slice CMR inputs, the latter being what can be acquired in real-time during an intervention. As shown in Fig. 2, *TetHeart* first constructs a patient-specific mesh representation from pre-operative data and then dynamically updates it during procedures using only the slices that can be acquired in real-time, down to a single one. *TetHeart* builds on deep marching tetrahedra Shen et al. (2021), which combines the optimization efficiency of signed distance fields with the geometric flexibility of tetrahedral meshes, providing a coherent space for joint shape–motion learning. Given this, our main contributions are as follows:

- We propose a slice-adaptive 2D–3D feature fusion mechanism that allows us to dynamically aggregate features from arbitrarily located slices, enabling robust reconstruction across varying input sparsity. We further introduce a full-to-sparse distillation strategy that transfers knowledge from dense to sparse regimes to ensure accurate reconstruction under extreme input sparsity.

- We develop a weakly supervised motion learning scheme that requires annotations only at key phases (end-diastolic and end-systolic) and learns motion through temporal consistency. This is essential in practice because it makes the training feasible and affordable across diverse clinical environments.

- We demonstrate that a single model trained once perform well across offline and online scenarios, achieving state-of-the-art results on ACDC Bernard et al. (2018), M&Ms Campello et al. (2021), and M&Ms-2 Martín-Isla et al. (2023). It also shows strong performance by further validating on a privately collected interventional CMR dataset and the public 4DM Yuan et al. (2023) dataset without retraining, demonstrating its potential for deployment in real-world clinical environments.

In short, the combination of our feature fusion and training scheme yields a novel unified framework that enables consistent reconstruction and motion inference across a wide spectrum of observational sparsity. As a result, *TetHeart* can operate both offline and online, enabling comprehensive, patient-specific motion tracking across the entire clinical workflow. We achieve state-of-the-art accuracy for offline use with full-stack CMR, and perhaps more importantly, we unlock online use in intra-procedural scenarios where only sparse observations are available, which cannot be handled using current methods. Code will be made public.

## 2 Related Work

We first review current methods to recovering static/dynamic heart, then the imaging modality we focus on.

### 2.1 Static Cardiac Shape Modeling

Reconstructing the heart from 3D medical images holds significant clinical value. However, CMR images often exhibit low through-plane resolution, making accurate reconstruction challenging. Current solutions Van Assen et al. (2006); Bai et al. (2015); Villard et al. (2018); Duan et al. (2019); Attar et al. (2019); Chen et al. (2021); Xia et al. (2022); Beetz et al. (2022); Hu et al. (2023); Xiao et al. (2024) use either traditional optimization-based techniques or deep learning-based methods.

**Optimization-Based Approaches** usually rely on a two-stage pipeline, often starting from contours manually extracted from the CMR images. For example, Villard et al. (2018) starts from a predefined tubular mesh that is deformed to minimize a contour-matching loss, and in Hu et al. (2023), a 3D active shape model and an intensity model are used to align initial meshes with GT contours. Even though this works, relying on test-time optimization makes the reconstruction time-consuming, taking from tens of seconds to several minutes per frame. When applied to a full image sequence, this may extend to over an hour, making it impractical for real-time applications. Moreover, these methods have only been evaluated using GT contours, requiring labor-intensive manual annotation.

**Deep Learning-Based Approaches** leverage prior knowledge learned from large datasets to accelerate mesh reconstruction. For instance, in Beetz et al. (2022), contours are treated as sparse point clouds and decoded into 3D meshes using point- and graph-convolutions. Similarly, in Chen et al. (2021), contours are embedded into a 3D volume, and a model composed of 3D CNNs and GCNs is used to progressively deform a template mesh. These methods are much faster but also have drawbacks. Reconstructing meshes solely from extracted contours or points fails to exploit the rich appearance information contained in cine images Xiao et al. (2024). Furthermore, as contour extraction is not necessarily perfect, such two-step approaches are subject to error accumulation Chen et al. (2021).

### 2.2 Dynamic Cardiac Motion Modeling

Arguably, when reconstructing deforming hearts from sequences of CMR scans, one could simply run a static reconstruction algorithm on each individual frame. However, this would fail to exploit temporal consistency and would make it difficult to establish inter-frame correspondences. Thus, several motion-based approaches that compute the deformation from frame to frame have been proposed.

When full CMR stacks are available, recent deep learning-based approaches Meng et al. (2022a;b; 2023); Yuan et al. (2023); Ye et al. (2023) can be used. MulViMotion Meng et al. (2022b) trains 3D CNNs to estimate left ventricular myocardial motion by predicting voxel-based deformation fields from CMR data and then building 3D meshes by warping segmentation masks. DeepMesh Meng et al. (2023) decouples mesh reconstruction from motion estimation. It uses 3D CNNs and interpolation to predict per-vertex deformation. 4DMR Yuan et al. (2023) models the myocardium using implicit representations. Decoupled shape and motion latent codes are optimized at test time to predict a source shape and motion fields. We use MulViMotion, DeepMesh, and 4DMR as baselines because they are among the best current methods.

Unfortunately, they also suffer from several limitations. First, some of them focus solely on the myocardium, neglecting other structures and thereby preventing comprehensive heart assessment. Second, optimization-based methods like 4DMR are time-consuming at inference time. Finally and most damagingly, they require

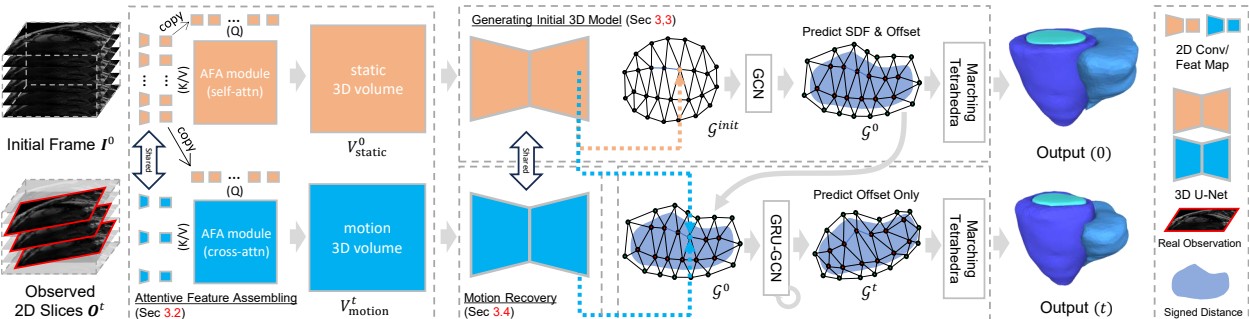

Figure 2: *TetHeart* reconstructs cardiac motion from observed 2D slices. At $t = 0$, a 3D CMR stack $\mathbf{I}^0$ is processed by the AFA module with self-attention to extract a volumetric static feature $V_{\text{static}}^0$, which is used to generate the initial tetrahedra $\mathcal{G}^0$. For each subsequent time step $t$, the observed 2D slices $\mathbf{O}^t$ are encoded via the AFA module with cross-attention to obtain a volumetric motion feature $V_{\text{motion}}^t$, which is used to recover motion by predicting offsets to deform $\mathcal{G}^0$. Weights are shared between static and dynamic branch.

full scans that cannot be acquired at a sufficient frame rate, which precludes intra-procedural use as discussed below.

## 2.3 Real-Time MR Imaging

CMR imaging has a reputation for being a "slow" modality. Acquiring a full MRI stack involves sequential slice acquisition over a 4–6 heartbeat breath-hold, which can take several minutes Nayak et al. (2022); Rogers et al. (2023). This is feasible for pre-interventional planning but impractical for intra-procedural use.

Recently, real-time MRI (RT-MRI) has gained favor due to its ability to capture dynamic processes without gating, synchronization, or repetition Nayak et al. (2022). Its applications include complex electrophysiology procedures Paetsch et al. (2019), congenital heart disease Hauptmann et al. (2019), and cardiac interventions Gao et al. (2012); Xu et al. (2014). RT-MRI provides excellent soft-tissue contrast for assessing interventions like ablations Bhagirath et al. (2015); Rogers et al. (2023), allows continuous device visualization and tracking, and is free of ionizing radiation Paetsch et al. (2019); Campbell-Washburn et al. (2017).

Although RT-MRI enables rapid continuous image acquisition, it still involves a fundamental tradeoff between spatial resolution, temporal resolution, artifacts and reconstruction latency Nayak et al. (2022). Therefore, achieving a sufficient temporal resolution (e.g., 20–25 Hz) for use during invasive procedures, can only be done at the cost of acquiring a limited number of 2D slices at a lower spatial resolution Gao et al. (2012); Xu et al. (2014); Campbell-Washburn et al. (2017); Nayak et al. (2022); Rogers et al. (2023). We are not aware of any current method that can reconstruct 3D cardiac motion using this kind of data. Yet, delivering a high-quality, dynamic 3D heart model to the intervention room based on real-time observations is crucial for guiding high-precision minimally invasive cardiac procedures Linte et al. (2009); Huang et al. (2009); Gao et al. (2012) and for real-time localization of devices within the heart Paetsch et al. (2019); Tampakis et al. (2023). This is the application that motivated us to begin this study.

## 3 Method

We now present *TetHeart*, a framework for 4D cardiac shape and motion reconstruction from either full CMR volumes or sparse 2D slice sets, as shown in Fig. 2. We first describe our heart representation based on deep marching tetrahedra Shen et al. (2021), an explicit–implicit hybrid model combining the optimization efficiency of signed distance fields with the geometric flexibility of tetrahedral meshes. Next, we introduce the Attentive 2D–3D Feature Assembler (AFA) that we design, which adaptively aggregates features from arbitrary numbers of 2D slices at varying spatial locations. Finally, we detail our weakly supervised training scheme, enabling effective motion learning from limited annotations available only at key cardiac phases.

### 3.1 Formalization

To model deforming hearts, we use the deep marching tetrahedra formalism. We discretize the 3D space using a deformable tetrahedral grid, in which each vertex possesses an signed distance field (SDF) value, denoted as $\mathcal{G} = (\{\mathbf{v}_i\}, \{s_i\}, \mathcal{T})$, where $\mathbf{v}_i$ and $s_i$ are the vertices and their corresponding SDFs in $\mathcal{T}$, the set of all tetrahedrons. The explicit surface can then be extracted using the differentiable marching tetrahedra algorithm Doi & Koide (1991).

Given a 3D heart model $\mathcal{G}^0$ reconstructed from the initial frame $\mathbf{I}^0 \in \mathbb{R}^{D \times H \times W}$, our goal is to estimate a deformed model $\mathcal{G}^t$ at each time instant $t$, based on subsequent observations $\mathbf{O}^t$ that reflect the ongoing cardiac motion. We do this by updating the vertices of $\mathcal{G}^0$:

$$\mathcal{G}^t = (\{\mathbf{v}_i^t\}, \{s_i\}, \mathcal{T}) \, , \mathbf{v}_i^t = \mathcal{D}(\mathbf{v}_i^0, \mathbf{O}^t) \, , \tag{1}$$

where $\mathbf{v}_i^0$ is a grid vertex from $\mathcal{G}^0$, and $\mathbf{v}_i^t$ its counterpart after deformation. $\mathcal{D}$ is the deformation model which we instantiate in the following sections to make it possible to recover deformations across the whole sequence.

The observations $\mathbf{O}^t$ can be obtained in two manners.

1. **Online.** For intra-procedural use, $\mathbf{I}^0 \in \mathbb{R}^{D \times H \times W}$ is a full-slice image acquired before the intervention, taking as much time as needed. However, it has to be possible to acquire $\mathbf{O}^t$ at sufficient frame rate and, consequently, it can only comprise a small set—from 1 to 3—of 2D slices at lower spatial resolution and at arbitrary locations Gao et al. (2012).

2. **Offline.** Full-slice CMR images are available both initially and at time $t$, obtained from a slow slice-by-slice scanning process. In this case, $\mathbf{O}^t$ and $\mathbf{I}^0$ are of the same dimension $\mathbb{R}^{D \times H \times W}$.

In the following sections, we first introduce the AFA module and then describe how it is incorporated into our approach to generate the initial shape model from $\mathbf{I}^0$ and then deform it given $\mathbf{O}^t$. Whether it is a full scan or a sparse set of slices, we use the same approach in both cases.

### 3.2 Attentive 2D-3D Feature Assembler

Because the observations can be either a full stack, a sparse set of slices, or anything in between, we need a mechanism to extract useful features in all cases. To this end, we developed the Attentive 2D–3D Feature Assembler (AFA) module that takes $\mathbf{I}^0$ and $\mathbf{O}^t$ as input and produces features.

Let us write $\mathbf{O}^t$ as $\{\mathbf{I}_s^t \in \mathbb{R}^{H \times W} | s = 1, \ldots, S\}$, a collection of 2D slices where $S$ is the number of slices. With this, $\mathbf{I}^0$ can be written as $\{\mathbf{I}_d^0 \in \mathbb{R}^{H \times W} | d = 1, \ldots, D\}$. A full-slice observation is simply one where $S = D$.

In practice, we first perform 2D convolutions on each individual slice to create a collection of 2D feature maps $F_{2d}^0$ and $F_{2d}^t$ from $\mathbf{I}^0$ and $\mathbf{O}^t$. Given that $\mathbf{O}^t$ is not necessarily in the same coordinate system as $\mathbf{I}^0$, we use the metadata associated in the DICOM header to position them spatially in the $\mathbf{I}^0$ volume. This links each 2D location in the feature maps to a precise 3D location. In the remainder of this section, we first discuss how we use these 2D feature maps to compute volumetric *motion features* from $F_{2d}^0$ and $F_{2d}^t$, which are obtained from slices acquired at different times and can be used to estimate deformations. We then describe how we derive *static features* from a dense set of slices all acquired at the same time, which can be used to reconstruct the initial static model.

**Motion Features.** A naive way to use attention to construct motion features would be to treat $F_{2d}^0$ as the query and $F_{2d}^t$ as both keys and values. However, this would be inefficient because global attention aims to capture long-range dependencies, which is not needed to model the inherently local nature of anatomical motion. Also, this requires computing full pairwise relationships between all query and key positions, which is computationally expensive when using many slices and high-resolution feature maps.

To avoid this, we take our inspiration from the Swin Transformer Liu et al. (2021) and use instead a variant of attention that preserves locality and significantly reduces computational complexity. For each voxel in

$F_{2d}^0$, we first identify $k_1$ spatially nearest 2D slices, or all slices if there are only $k_1$ or less. Within each selected slice, we then retrieve the features at the $k_2$ spatially closest positions to serve as keys and values. Attention is subsequently computed only between the query and its corresponding set of localized keys and values. This can be written as

$$V_{\text{motion}}^t = AFA(F_{2d}^0, F_{2d}^t) = \text{MHAttention}(Q, K, V) \text{, where } Q = F_{2d}^0, \ K = V = \text{NN-Select}_{F_{2d}^0, k_1, k_2}(F_{2d}^t) \text{,}$$

where *NN-Select* denotes the nearest-slice and position-selecting operation. *MHAttention* denotes a Multi-Head Attention Vaswani et al. (2017) operation. A learnable positional embedding is used to encode spatial information. In practice, we set $k_1 = 3$ and $k_2 = 9$. This ensures that the computational complexity of our attention mechanism is approximately equivalent to that of a 3D convolution with a kernel size of 3. As a result, the AFA module achieves high computational efficiency while retaining the flexibility and adaptability of attention-based feature aggregation. The complete algorithm pipeline is described in Sec. B.

While our AFA module effectively encodes features from varying 2D slice configurations, a persistent challenge remains in the online scenario, where the limited number of slices can hinder accurate 3D motion reconstruction. To alleviate this issue, we introduce a distillation loss that encourages the model to transfer knowledge from full-slice inputs to partial-slice inputs during training. In short, we predict two versions of motion features from full-slice observation and its randomly sampled slice subset. The distillation loss encourages the encoded feature from partial slices to be close to the feature from full-slice input. We formulate the loss we minimize to achieve this in Section 3.5.

**Static Features.** The motion features described above relate $\mathbf{I}^0$ to $\mathbf{O}^t$. The same mechanism can also be used to relate $\mathbf{I}^0$ to itself and produce features that can be used for static reconstruction. This can be similarly written as

$$V_{\text{static}}^0 = AFA(F_{2d}^0, F_{2d}^0) = \text{MHAttention}(Q, K, V) \text{, where } Q = F_{2d}^0, \ K = V = \text{NN-Select}_{F_{2d}^0, k_1, k_2}(F_{2d}^0) \text{.}$$

**Generating Final Features.** After obtaining the volumetric static and motion features $V_{\text{static}}^0$ and $V_{\text{motion}}^t$, we use a U-Net-like Ronneberger et al. (2015); Isensee et al. (2021) architecture for further encoding. Just as standard U-Net architecture, it also consists of a downsampling and an upsampling stream. The last level features of the upsample stream are taken as the final features, which we denoted as $F_{\text{static}}^0$ and $F_{\text{motion}}^t$.

Note that the initial 2D encoding block and subsequent 3D convolutions are shared for static and motion feature extraction. This unified design allows motion prediction to benefit from the rich 3D spatial information learned in the static reconstruction task, facilitating more accurate motion estimation—an advantage overlooked by some previous works Meng et al. (2023); Yuan et al. (2023). As shown in the experiments, this design leads to superior motion reconstruction performance.

### 3.3 Initial Heart Model Reconstruction

Given $F_{\text{static}}^0$ computed as discussed above, our first task is to reconstruct the initial tetrahedral mesh $\mathcal{G}^0$. To this end, we start from tetrahedral grid $\mathcal{G}^{init}$ obtained by uniformly sampling a unit cube. We then trilinearly interpolate $F_{\text{static}}^0$ at the vertices of $\mathcal{G}^{init}$. An SDF value and an offset for each vertex are predicted using a GCN to create $\mathcal{G}^0$. We write

$$\begin{aligned}
\mathcal{G}^0 &= (\{\mathbf{v}_i'\}, \{s_i\}, T), \\
\mathbf{v}_i' &= \mathbf{v}_i + \Delta\mathbf{v}_i, \\
(\Delta\mathbf{v}_i, s_i) &= \text{GCN}([F_{\text{static}}^0(\mathbf{v}_i), \mathbf{v}_i]) \text{,}
\end{aligned} \tag{2}$$

where $\mathbf{v}_i$ is a grid vertex of $\mathcal{G}^{init}$ and $\mathbf{v}_i'$ the same grid vertex after deformation. $F_{\text{static}}^0(\mathbf{v})$ denotes feature extraction at $\mathbf{v}$ from $F_{\text{static}}^0$ with trilinear interpolation, and $[\cdot, \cdot]$ denotes concatenation. To enhance spatial sensitivity, the vertex location is appended to the input before it is passed to the GCN. For datasets with $C$ classes, we set the output channel to $C \times 4$, 3 for deformation and 1 for SDF. Following prior works Wickramasinghe et al. (2020); Kong et al. (2021); Bongratz et al. (2022); Cruz et al. (2021), our network also has an auxiliary segmentation prediction branch during training.

### 3.4 Motion Recovery

The static reconstruction scheme described above delivers comparable or even better reconstruction performance than approaches based on pure 3D U-Net architectures Wickramasinghe et al. (2020); Kong et al. (2021); Bongratz et al. (2022); Cruz et al. (2021), with the added benefit that it also allows the independent encoding of individual 2D slices. We now exploit this mechanism to compute deformations for the initial static shape from sets of slices acquired later. Given $\mathcal{G}^0$, $F_{\text{static}}^0$, and $F_{\text{motion}}^t$ computed as described above, we use a combination of GCN and GRU Cho et al. (2014) layers to effectively aggregate spatial information and instantiate the deformation model $\mathcal{D}$ of Eq. 1. We write

$$F^{cat}(\mathbf{v}_i^{(s)}) = [F_{\text{static}}^0(\mathbf{v}_i^{(s)}), F_{\text{motion}}^t(\mathbf{v}_i^{(s)})] \,, \tag{3}$$

$$F^{gcn}(\mathbf{v}_i^{(s)}) = \text{GCN}(F^{cat}(\mathbf{v}_i^{(s)})) \,, \tag{4}$$

$$\mathbf{h}_i^{(s+1)} = \text{GRU}([F^{gcn}(\mathbf{v}_i^{(s)}), \mathbf{v}_i^{(s)}], \mathbf{h}_i^{(s)}) \,, \tag{5}$$

$$\mathbf{v}_i^{(s+1)} = \mathbf{v}_i^{(s)} + \text{MLP}(\mathbf{v}_i^{(s)}, \mathbf{h}_i^{(s+1)}) \,. \tag{6}$$

In Eq. 3, the vertex features are extracted by trilinear interpolating $F_{\text{static}}^0$ and $F_{\text{motion}}^t$ at their locations. They are then concatenated and passed through Eq. 4-6 to gradually update the vertex position. Eqs. 3-6 are repeated twice.

### 3.5 Two-Stage Weakly Supervised Training Using Only Keyframe Annotations

Creating high-quality annotations for an entire sequence is costly. Therefore, many datasets provide annotations only for keyframes, such as end-diastolic (ED) and end-systolic (ES) phases. This rules out full supervision. To address this, we propose a two-stage weakly supervised pipeline, which makes training the reconstruction and motion branches feasible using only the available keyframe annotations. The core idea is to first train a static reconstruction branch on labeled frames, and then repurpose it to initialize shapes for unlabeled frames, from which motion dynamics can be recovered. The pipeline is illustrated in Fig. 7.

**Training the Reconstruction Branch.** We start by training the reconstruction branch of Section 3.3. At each training iteration, we randomly select a labeled image $\mathbf{I}$, that is, an ED or ES frame, with ground truth segmentation and mesh annotation $L^{gt}, \mathcal{M}^{gt}$. Let $L^p, \mathcal{G}^p$ be our model's segmentation and tetrahedra prediction. We minimize the loss

$$\mathcal{L}_{shape}(\mathbf{I}) = \lambda_{cd}\mathcal{L}_{cd}(\mathbf{MT}(\mathcal{G}^p), \mathcal{M}^{gt}) + \lambda_{sdf}\mathcal{L}_{sdf}(\mathcal{G}^p, \mathcal{M}^{gt}) + \lambda_{ce}\mathcal{L}_{ce}(L^p, L^{gt}) \,,$$

where $\mathcal{L}_{cd}$ is the chamfer distance, $\mathcal{L}_{sdf}$ is an L1-loss used to supervise the predicted SDF values at tetrahedral grid vertices with the SDF values queried from the ground-truth mesh. $\mathcal{L}_{ce}$ is the cross-entropy loss defined on the segmentation map. $\mathbf{MT}$ denotes the marching tetrahedra algorithm Doi & Koide (1991) used to convert $\mathcal{G}^p$ into surface mesh. $\lambda_{cd}, \lambda_{sdf}, \lambda_{ce}$ are set to $1.0, 0.1, 0.1$. The data is diverse enough for our model to generalize to arbitrary timesteps even though annotations are only available for ED/ES frames.

**Training the Motion Branch.** We then train the motion branch. At each iteration, we randomly select a sequence $\{\mathbf{I}^t\}_{t=0}^T$, and choose an unlabeled frame $\mathbf{I}^u$ and a labeled frame $\mathbf{I}^l$ from it. We first predict $\mathcal{G}^u$ from $\mathbf{I}^u$ using the trained reconstruction branch. Next, assuming that $\mathbf{I}^l$ comprises a total of $D$ slices, we randomly pick 1 to $D$ 2D slices to form $\hat{\mathbf{I}}^l$ to simulate the online few-slice case. Crucially, $\mathbf{I}^l$ and $\hat{\mathbf{I}}^l$ undergo random downsampling/rotation/noise jittering to simulate image quality degradation encountered in the interventional setting. The corresponding ground-truth mesh $\mathcal{M}^l$ undergoes the same transformation. $V_{\text{motion}}^l$ and $\hat{V}_{\text{motion}}^l$ are generated from $\mathbf{I}^l$ and $\hat{\mathbf{I}}^l$ using the AFA module. We write

$$V_{\text{motion}}^l = AFA(F_{2d}^u, F_{2d}^l), \quad \hat{V}_{\text{motion}}^l = AFA(F_{2d}^u, \hat{F}_{2d}^l) \,.$$

We use these 3D features to generate two versions of deformed tetrahedra using Eq. 1: $\mathcal{G}^{u \to l}$ from full-slice encoded feature $V_{\text{motion}}^l$, and $\hat{\mathcal{G}}^{u \to l}$ from few-slice encoded feature $\hat{V}_{\text{motion}}^l$. The network is trained with

$$\mathcal{L}_{motion}(\mathbf{I}^l, \mathbf{I}^u) = \mathcal{L}_{distill}(V_{\text{motion}}^l, \hat{V}_{\text{motion}}^l) + \mathcal{L}_{cd}(\mathbf{MT}(\mathcal{G}^{u \to l}), \mathcal{M}^l) + \mathcal{L}_{cd}(\mathbf{MT}(\hat{\mathcal{G}}^{u \to l}), \mathcal{M}^l),$$

$$\mathcal{L}_{distill}(V_{\text{motion}}^l, \hat{V}_{\text{motion}}^l) = ||\hat{V}_{\text{motion}}^l - \mathtt{sg}(V_{\text{motion}}^l)||_2 \,,$$

where `sg` denotes the stop-gradient operation. Minimizing the *distillation loss* $\mathcal{L}_{distill}$ encourages features from few-slice input $\hat{\mathbf{I}}^l$ to align with those from the full-slice input $\mathbf{I}^l$, while minimizing the chamfer losses enables the deformation model to infer accurate motion from the feature maps encoded from any number of 2D slices, while retaining the ability to infer motion from complete 3D full-slice feature map. As a result, minimizing $\mathcal{L}_{motion}$ guarantees good reconstruction results for both full and few-slice inputs. This training strategy allows training a 4D model without dense temporal annotations.

Once trained, our model can be used for new patients without retraining. Note that our training strategy actually enables the model to predict cardiac deformation starting from any phase, not just the ED phase. Additionally, by employing an aggressive sampling and augmentation scheme, the model learns to reconstruct motion from a highly diverse set of input configurations. As a result, even though the online scenario is simulated during training, our model shows strong extendibility. It can robustly handle cases where $\mathbf{O}^t$ originates from a different sequence or where there is a scanning angle discrepancy relative to $\mathbf{I}^0$, consistently producing reliable motion predictions despite these challenges, as will be shown in the experiment section.

## 4 Experiments

### 4.1 Experimental Settings

**Datasets and Metrics.** We train and validate our method on a unified dataset constructed from three large and diverse publicly available datasets: *ACDC*, *M&Ms*, and *M&Ms-2* Bernard et al. (2018); Campello et al. (2021); Martín-Isla et al. (2023). Segmentation for the left ventricle (LV), right ventricle (RV), and myocardium (MYO) are provided, but only for the ED and ES frames. The unified dataset comprises 835 CMR sequences. To prevent data leakage, we use a subject-level split to randomly split it into training, validation, and testing sets in a ratio of 60%, 20%, and 20%. Dataset details are provided in Appendix A.1.

We use two additional datasets for external evaluation in a clinical context where model retraining is not feasible. The first dataset, referred to as *iCMR*, was collected on the interventional MRI system depicted in Fig. 1(c) under both rest and elevated-heart-rate conditions. It is designed with the target online scenario in mind. For each subject, full short-axis cine stacks were acquired at rest, while additional single mid-ventricular slices were collected at rest and during mild in-scanner exercise to simulate heart-rate variations. The dataset thus reflects realistic cardiac motion dynamics encountered in online interventional MRI. Details on the scanning protocol, equipment, and exercise procedure are provided in Appendix A.2. To evaluate full-sequence performance across the entire cardiac cycle beyond the keyframes, we use the second external dataset, *4DM* Yuan et al. (2023), which focuses on left ventricular myocardium reconstruction and provides mesh annotations for every frame. This dataset allows rigorous evaluation of the proposed weakly supervised motion learning scheme.

For evaluation, we use the L2-norm Chamfer Distance (CD) to measure 3D reconstruction quality and the Dice score to quantify intra-slice segmentation results. We also report the Mean-Absolute Error of LV/RV ES Volume (LVESV/RVESV), LV/RV ejection fraction (LVEF/RVEF) derived from our predicted meshes. LV/RVEF are representative cardiac function indices used for diagnosis. See Appendix A.3 for details. We also perform statistical significance tests in Appendix D.

**Implementation Details.** In all experiments, the slices are resampled by linear interpolation to a spacing of $1.25 \times 1.25$ mm. From this, the AFA module constructs a 3D volume with an effective spacing of $1.25 \times 1.25 \times 2$ mm. See Appendix B for model and optimization details.

**Baselines.** We compare our method against several existing approaches for 4D shape reconstruction. For the online interventional setting, as noted earlier, most methods assume access to a complete CMR stack, making them unsuitable for few-slice inputs. Therefore, we adapted several baselines to operate under sparse inputs, including modified versions of **4DMR** Yuan et al. (2023), **MR-Net** Chen et al. (2021), **MulViMotion** Meng et al. (2022b), and **DeepMesh** Meng et al. (2023). We also include two variants derived from shape reconstruction methods: MeshDeformNet Kong et al. (2021) and DeepCSR Cruz et al. (2021), denoted as **Ours-Mesh** and **Ours-SDF**, since their main differences from our method lie in the shape representation. For the traditional offline setting, in addition to the above methods, we further compare against Free-Form

| | | | | 1-Slice | | | | | | | | | | | | | | | | | |
|---|---|---|---|---|---|---|---|---|---|---|---|---|---|---|---|---|---|---|---|---|---|
| | | | | ACDC | | | | | | M&Ms | | | | | | M&Ms-2 | | | | | |
| Method | GT-Free | Optim-Free | Repre. | CD (mm²) ↓ | | | Dice (%) ↑ | | | CD (mm²) ↓ | | | Dice (%) ↑ | | | CD (mm²) ↓ | | | Dice (%) ↑ | | |
| | | | | Myo | LV | RV | Myo | LV | RV | Myo | LV | RV | Myo | LV | RV | Myo | LV | RV | Myo | LV | RV |
| 4DMR | ✗ | ✗ | SDF | 29.53 | - | - | 75.63 | - | - | 19.87 | - | - | 76.12 | - | - | 19.37 | - | - | 74.11 | - | - |
| MR-Net | ✗ | ✓ | Mesh | 28.61 | 40.70 | 46.15 | 77.43 | 82.12 | 73.33 | 14.59 | 16.46 | 25.11 | 77.59 | 80.35 | 73.94 | 15.61 | 16.68 | 25.23 | 76.29 | 82.66 | 76.23 |
| MulViMotion | ✓ | ✓ | Mesh | 29.13 | - | - | 75.02 | - | - | 18.43 | - | - | 72.12 | - | - | 19.51 | - | - | 70.21 | - | - |
| DeepMesh | ✓ | ✓ | Mesh | 27.47 | - | - | 76.89 | - | - | 14.29 | - | - | 76.15 | - | - | 15.35 | - | - | 73.75 | - | - |
| Ours-Mesh | ✓ | ✓ | Mesh | 17.42 | 25.11 | 42.33 | 80.14 | 84.90 | 74.32 | 11.26 | 14.17 | 23.19 | 79.88 | 82.11 | 75.96 | 11.98 | 15.11 | 22.63 | 77.66 | 84.97 | 77.99 |
| Ours-SDF | ✓ | ✓ | SDF | 18.65 | 26.45 | 43.78 | 79.55 | 83.43 | 74.00 | 12.16 | 14.88 | 23.91 | 79.09 | 81.62 | 74.23 | 12.15 | 15.72 | 23.34 | 77.01 | 84.37 | 77.15 |
| Ours | ✓ | ✓ | Tet | **15.24** | **23.65** | **40.64** | **82.25** | **86.20** | **75.42** | **9.76** | **12.37** | **21.42** | **81.00** | **84.45** | **77.32** | **10.12** | **13.89** | **20.27** | **79.33** | **86.06** | **79.65** |
| | | | | 5-Slice | | | | | | | | | | | | | | | | | |
| | | | | ACDC | | | | | | M&Ms | | | | | | M&Ms-2 | | | | | |
| Method | GT-Free | Optim-Free | Repre. | CD (mm²) ↓ | | | Dice (%) ↑ | | | CD (mm²) ↓ | | | Dice (%) ↑ | | | CD (mm²) ↓ | | | Dice (%) ↑ | | |
| | | | | Myo | LV | RV | Myo | LV | RV | Myo | LV | RV | Myo | LV | RV | Myo | LV | RV | Myo | LV | RV |
| 4DMR | ✗ | ✗ | SDF | 24.56 | - | - | 77.03 | - | - | 16.16 | - | - | 77.33 | - | - | 15.52 | - | - | 76.67 | - | - |
| MR-Net | ✗ | ✓ | Mesh | 18.82 | 24.22 | 36.54 | 79.54 | 84.25 | 74.67 | 10.60 | 12.16 | 20.28 | 78.70 | 83.31 | 76.91 | 10.76 | 12.05 | 19.25 | 77.84 | 86.03 | 78.79 |
| MulViMotion | ✓ | ✓ | Mesh | 23.05 | - | - | 77.03 | - | - | 15.55 | - | - | 74.03 | - | - | 15.03 | - | - | 73.42 | - | - |
| DeepMesh | ✓ | ✓ | Mesh | 18.34 | - | - | 78.05 | - | - | 11.19 | - | - | 77.59 | - | - | 10.98 | - | - | 75.33 | - | - |
| Ours-Mesh | ✓ | ✓ | Mesh | 11.66 | 19.39 | 33.91 | 82.92 | 85.88 | 75.73 | 9.14 | 11.07 | 18.62 | 80.99 | 85.11 | 78.67 | 8.68 | 10.96 | 16.82 | 79.82 | 87.99 | 80.13 |
| Ours-SDF | ✓ | ✓ | SDF | 12.21 | 20.03 | 34.52 | 82.35 | 85.32 | 75.11 | 9.63 | 11.69 | 19.26 | 80.45 | 84.56 | 78.05 | 9.22 | 11.55 | 17.42 | 79.21 | 87.41 | 79.53 |
| Ours | ✓ | ✓ | Tet | **10.22** | **18.31** | **32.61** | **84.38** | **87.32** | **77.23** | **7.93** | **9.72** | **17.39** | **82.83** | **86.76** | **79.27** | **7.30** | **9.96** | **15.58** | **81.24** | **89.37** | **81.48** |

Table 1: *Few-slice quantitative evaluation on unified dataset.* We compared the predicted 3D meshes by deforming from ED frame to ES frame to the ground-truth 3D meshes. **1- / 5-Slice**: Number of slices used to recover the motion. GT-Free: no need of ED GT mesh at inference time to predict motion. Optim-Free: no need of test-time optimization.

| Method | LVESV MAE (ml)↓ | LVEF MAE (%) ↓ | RVESV MAE (ml)↓ | RVEF MAE (%) ↓ |
|---|---|---|---|---|
| | 1-Slice | | | |
| MR-Net Chen et al. (2021) | 9.76 | 7.87 | 16.40 | 11.75 |
| Ours-Mesh Kong et al. (2021) | 8.92 | 6.37 | 16.41 | 9.83 |
| Ours-SDF Cruz et al. (2021) | 9.35 | 6.82 | 17.04 | 10.21 |
| Ours | **7.51** | **4.83** | **15.03** | **8.51** |
| | 5-Slice | | | |
| MR-Net Chen et al. (2021) | 7.34 | 6.29 | 13.45 | 8.87 |
| Ours-Mesh Kong et al. (2021) | 7.64 | 5.11 | 12.81 | 7.33 |
| Ours-SDF Cruz et al. (2021) | 7.93 | 5.32 | 13.12 | 7.59 |
| Ours | **6.86** | **4.30** | **11.88** | **6.55** |
| | Full-Slice | | | |
| FFD Rueckert et al. (2002) | 23.53 | 14.95 | 24.06 | 15.87 |
| dDemons Vercauteren et al. (2007) | 33.97 | 22.11 | 63.03 | 38.52 |
| MR-Net Chen et al. (2021) | 6.75 | 4.16 | 12.05 | 7.01 |
| Ours-Mesh Kong et al. (2021) | 5.07 | 2.23 | 9.85 | 4.96 |
| Ours-SDF Cruz et al. (2021) | 5.22 | 2.34 | 9.98 | 5.01 |
| Ours | **4.84** | **2.04** | **9.65** | **4.77** |

Table 2: *Clinical indexes evaluation on M&Ms dataset.* 1- / 5- / Full-Slice denotes the number of slices used to recover motion.

Deformation (**FFD**) Rueckert et al. (2002), diffeomorphic Demons (**dDemons**) Vercauteren et al. (2007), and **MR-Net w. nnU-Net**. Details about the modified baselines are provided in Appendix C.

## 4.2  Motion Estimation from a Few Slices

We present results under the online few-slice scenario that represents the primary motivation of this work. Fig. 3 shows a representative motion reconstruction using only the central slice as input: one from a normal (NOR) subject and another from a case of dilated cardiomyopathy (DCM). Due to stronger contractile function, the normal heart exhibits larger deformations, making motion prediction inherently more challenging, as reflected by higher reconstruction errors. Nevertheless, our method delivers accurate predictions in both cases. Moreover, segmentation results on slices near the apex and base demonstrate that our model produces reliable deformations even in regions distant from the input slice. Complete reconstructed sequences and their corresponding volume–time curves are shown in Fig. 4.

Quantitative results on the full test set are reported in Tab. 1 and the top section of Tab. 2, including Chamfer Distance, Dice scores, and cardiac function indices. For consistency, all experiments were performed using the central 1 or 5 slices, though slices at arbitrary positions could have been used, as discussed in Section 4.4. *TetHeart* consistently outperforms all baselines without requiring ground-truth contours or segmentations as input at inference time. It infers cardiac motion directly from raw CMR images, eliminating the need for time-consuming test-time optimization, and achieves an inference speed of 12 FPS on an V100 GPU. Further acceleration could be achieved through model quantization or lightweight encoders to meet strict real-time constraints. In terms of model size, ours has 23M parameters, while the best competitor MR-Net has 22M parameters but performs worse and runs at similar speed. Notably, even when provided with a single slice, *TetHeart* produces meaningful physiological indices, underscoring its suitability for clinical deployment.

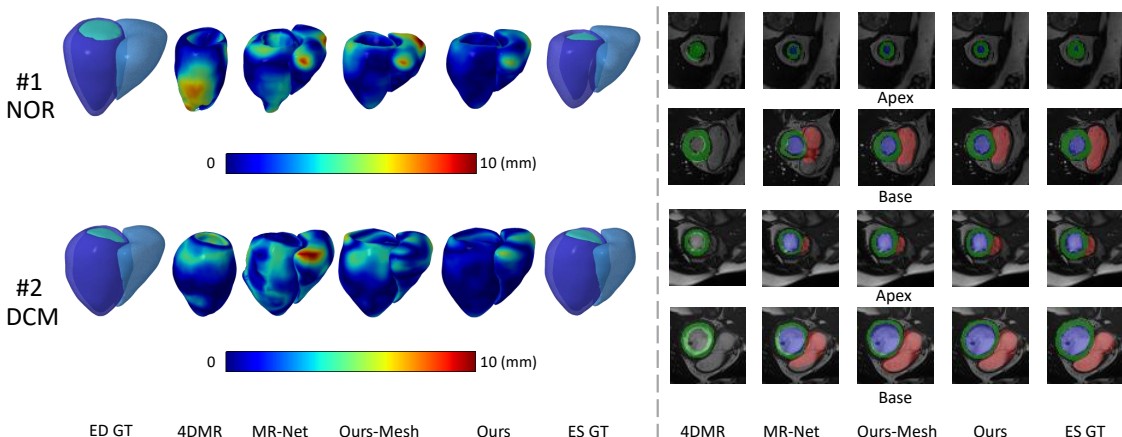

Figure 3: *1-Slice Comparison.* (Left) We predict meshes by deforming from ED to ES frame. Color indicates the magnitude of point-to-surface error. (Right) Segmentation results on slices at the apex and base location. Note 4DMR can only output motion for myocardium.

| Method | GT-Free | Optim-Free | Repre. | ACDC CD (mm²) ↓ Myo | LV | RV | ACDC Dice (%) ↑ Myo | LV | RV | M&Ms CD (mm²) ↓ Myo | LV | RV | M&Ms Dice (%) ↑ Myo | LV | RV | M&Ms-2 CD (mm²) ↓ Myo | LV | RV | M&Ms-2 Dice (%) ↑ Myo | LV | RV |
|---|---|---|---|---|---|---|---|---|---|---|---|---|---|---|---|---|---|---|---|---|---|
| FFD | ✗ | ✗ | Mesh | 21.84 | 38.85 | 51.04 | 69.46 | 79.56 | 72.04 | 42.01 | 63.23 | 88.77 | 62.84 | 76.65 | 66.23 | 40.79 | 66.42 | 95.66 | 63.13 | 79.80 | 76.02 |
| dDemons | ✗ | ✗ | Mesh | 15.72 | 22.97 | 35.85 | 77.06 | 86.87 | 73.60 | 32.16 | 44.39 | 61.15 | 65.83 | 79.87 | 65.09 | 34.08 | 48.77 | 66.22 | 67.25 | 83.44 | 72.39 |
| 4DMR | ✗ | ✗ | SDF | 10.89 | - | - | 80.03 | - | - | 8.42 | - | - | 82.33 | - | - | 8.25 | - | - | 82.67 | - | - |
| MR-Net | ✗ | ✓ | Mesh | 14.02 | 19.21 | 24.07 | 80.02 | 86.35 | 78.93 | 8.86 | 10.03 | 16.86 | 79.40 | 86.90 | 79.73 | 8.39 | 7.24 | 12.99 | 80.72 | 87.86 | 80.60 |
| MR-Net w. nnU-Net | ✓ | ✓ | Mesh | 15.11 | 20.45 | 26.13 | 79.66 | 85.38 | 76.45 | 9.88 | 10.25 | 18.60 | 78.56 | 85.00 | 77.21 | 9.83 | 9.24 | 13.35 | 79.56 | 85.89 | 78.31 |
| MulViMotion | ✓ | ✓ | Mesh | 17.21 | - | - | 78.10 | - | - | 13.75 | - | - | 75.31 | - | - | 12.33 | - | - | 75.23 | - | - |
| DeepMesh | ✓ | ✓ | Mesh | 16.42 | - | - | 79.89 | - | - | 9.24 | - | - | 78.51 | - | - | 9.20 | - | - | 77.75 | - | - |
| Ours-Mesh | ✓ | ✓ | Mesh | 6.80 | 11.84 | 23.41 | 86.62 | 88.11 | 84.74 | 4.26 | 5.08 | 13.92 | 84.09 | 87.12 | 83.73 | 4.15 | 5.12 | 11.04 | 85.01 | 88.67 | 86.65 |
| Ours-SDF | ✓ | ✓ | SDF | 7.21 | 12.65 | 22.78 | 86.95 | 88.43 | 84.00 | 4.44 | 5.26 | 14.21 | 83.87 | 86.91 | 83.77 | 4.30 | 5.44 | 11.32 | 84.55 | 88.37 | 86.10 |
| Ours | ✓ | ✓ | Tet | **6.63** | **11.51** | **22.15** | **87.12** | **88.96** | **84.21** | **4.11** | **4.85** | **13.68** | **84.31** | **87.43** | **83.99** | **4.02** | **5.04** | **10.82** | **85.33** | **89.13** | **86.94** |

Table 3: *Full-slice quantitative evaluation on unified dataset.* We compared the predicted 3D meshes by deforming from ED frame to ES frame to the ground-truth 3D meshes. GT-Free: no need of ED GT mesh at inference time to predict motion. Optim-Free: no need of test-time optimization.

Although Ours-Mesh and Ours-SDF incorporate the same sparse feature handling strategy as our full model, *TetHeart* still achieves superior performance. This confirms that the tetrahedral representation effectively preserves spatial coherence and serves as a more expressive basis for 3D motion inference under sparse observations. Fig. 5 illustrates performance as a function of slice count: the fewer slices used, the larger the gap between *TetHeart* and the ablated variants. Nevertheless, both variants still outperform all prior methods by a clear margin, demonstrating that the AFA module and the training strategy form a robust and generalizable solution for incomplete observations.

The relatively poor performance of 4DMR can be attributed to its simple MLP architecture, which may lack sufficient representational capacity to generalize well to the large and heterogeneous datasets considered here. Moreover, its test-time optimization strategy does not generalize well to severely under-sampled inputs. The weaker performance of MR-Net likely arises from the lack of reconstruction features and its sole reliance on contour points for deformation prediction, which overlooks rich cine image information.

## 4.3 Motion Estimation from Full Stacks

Since we had to modify existing techniques to compare them to ours in the few-slice scenario, we now compare our method to the original baselines on the full set of slices they were designed to operate on and report the results in Tab. 3 and the bottom of Tab. 2. In Fig. 4 and Fig. 8, we show qualitative results on the same hearts as in Fig. 3, but now using all slices as input instead of only one.

Remarkably, although the main driver of this work was operating on as few slices as possible, we also consistently outperform the baselines in the full-slice setting. We attribute this to several factors. Using predicted

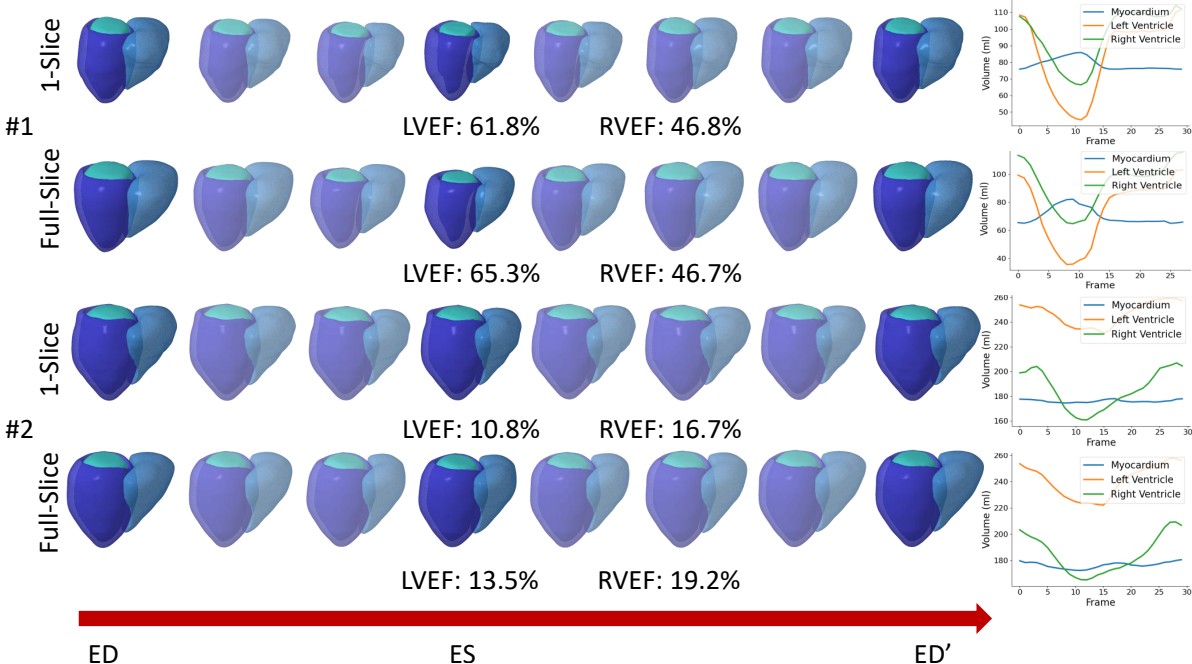

Figure 4: *Motion sequence prediction results using 1-/Full-slices by deforming from ED frame to the target frame.* Corresponding Volume-frame curve is given on the right. #1: NOR, #2: DCM.

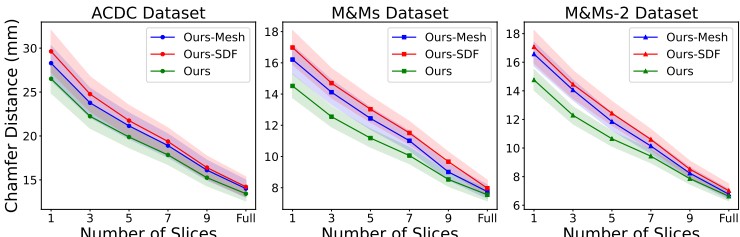

Figure 5: Chamfer distance as a function of the number of slices used to capture the motion.

contours in MR-Net leads to slightly worse performance, indicating that such a two-stage automated pipeline is prone to error accumulation, consistent with the conclusions in the original paper Chen et al. (2021). MulViMotion estimates voxel-wise motion fields and reconstructs meshes from warped segmentation maps, which is inherently less accurate than directly modeling meshes. Compared to DeepMesh, our model benefits from a shared encoding network, allowing the motion branch to leverage features from the reconstruction branch. Moreover, the AFA module dynamically aggregates spatially adjacent features, improving prediction quality.

## 4.4 Evaluation on External Datasets without Retraining

| Method | Training Dataset | | | Dice (%) ↑ | | |
|---|---|---|---|---|---|---|
| | ACDC | M&Ms | M&Ms-2 | Myo | LV | RV |
| 4DMR | ✓ | ✓ | ✓ | 71.33 | - | - |
| MR-Net | ✓ | ✓ | ✓ | 76.11 | 78.65 | 72.11 |
| Ours-Mesh | ✓ | ✓ | ✓ | 79.17 | 82.02 | 78.97 |
| Ours-SDF | ✓ | ✓ | ✓ | 78.42 | 81.11 | 78.21 |
| Ours | ✓ | - | - | 69.12 | 71.33 | 69.81 |
| | - | ✓ | - | 76.99 | 78.17 | 77.68 |
| | - | - | ✓ | 76.13 | 79.67 | 78.21 |
| | ✓ | ✓ | ✓ | **80.15** | **83.22** | **79.76** |

Table 4: *Evaluation results on the iCMR dataset.* We report the Dice Score as the evaluation metric. Check marks denote the dataset used for training.

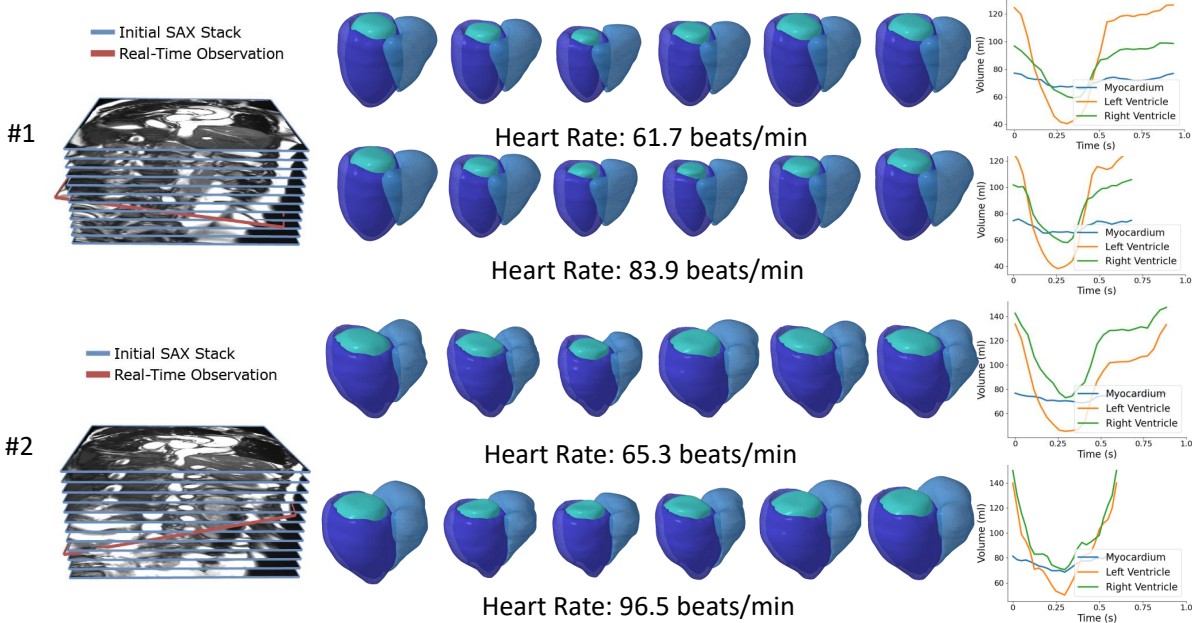

Figure 6: *Motion sequence prediction on the iCMR dataset.* (Left) Spatial configuration of the initial SAX stack and the online observations. (Middle) Cardiac motion sequences for each subject under rest and exercise conditions. (Right) Volume–time curves.

We now perform external evaluation on the iCMR and 4DM datasets *without* retraining. In Fig. 6, we show sequence predictions on the iCMR dataset, with the spatial positions of the real-time captured 2D slices visualized. As can be seen, *TetHeart* reconstructs plausible heart motion across the cardiac cycle under both rest and exercise conditions. Moreover, the predicted volume–time curves confirm the physiological plausibility of our motion reconstruction. Under the exercise condition, the heart exhibits a shorter ES–ED recovery period and a reduced duration for each cardiac cycle, which aligns well with expectations. The quantitative results in Tab. 4 also demonstrate that our model achieves the best performance. 4DMR exhibits poor generalization, likely due to significant distribution shifts across datasets. While MR-Net shows some generalization ability, it still performs worse than our method.

Moreover, training with a larger and more diverse dataset improves generalization. And consistent with the findings from previous evaluations, representing the heart as tetrahedra enables more effective spatial information retention and exchange, particularly in scenarios with limited slice input.

Please refer to Appendix E for more details and the external evaluation results on the 4DM dataset.

## 4.5   Ablation Studies

We conduct ablation studies to evaluate our design choices in Tab. 5. We first verify the effectiveness of the training strategy proposed in Sec. 3.5, especially the influence of slice subsampling and random downsampling/rotation/noise jittering when training the motion branch. To quantitatively evaluate this, we simulate interventional artifacts on the Unified dataset by applying down-sampling, noise jitter, and 10° rotation to ES frames and their GT meshes during testing. Tab. 5(left) confirms that our training strategy overcomes such challenges that are representative of those occurring in interventional scenarios. The clean ES results from Tab. 1 are provided for reference.

Next, we ablate the model design in Tab. 5(right). First, we examine the AFA module. Simply stacking 2D features yields similar performance to AFA in the full-slice setting but fails to generalize when the number or positions of slices vary, highlighting the need for a flexible aggregation mechanism. Removing positional encodings moderately degrades performance, confirming the importance of spatial information, while using only positional encodings leads to a substantial drop, showing that both image features and spatial coordinates are essential for effective motion prediction. When the motion branch is trained without parameter sharing and randomly initialized, performance decreases significantly, demonstrating that our shared-parameter design effectively transfers 3D knowledge from reconstruction to motion modeling.

| Method (1-Slice) | M&Ms CD (mm$^2$) ↓ | | |
|---|---|---|---|
| | Myo | LV | RV |
| Ours w.o. Sec. 3.5 aug | 14.45 | 18.72 | 30.67 |
| Ours | 10.98 | 13.76 | 23.66 |
| Ours (clean input) | 9.76 | 12.37 | 21.42 |

| Method | M&Ms Myo CD (mm$^2$) ↓ | | |
|---|---|---|---|
| | 1-Slice | 5-Slice | Full-Slice |
| Ours | 9.76 | 7.93 | 4.11 |
| - AFA module | - | - | 4.22 |
| - position embedding | 11.23 | 9.04 | 4.67 |
| - image feature in query $Q$ | 16.67 | 12.37 | 6.44 |
| - shared encoding network | 20.66 | 11.63 | 6.43 |
| - distillation loss | 10.34 | 8.12 | 4.03 |

Table 5: *Ablation study.* (left) Evaluation results with simulated interventional artifacts. We compared the predicted meshes by deforming from ED to interventional-like ES. (right) Ablation results on M&Ms dataset.

Lastly, we ablate the impact of the distillation loss. Without it, the model performs slightly better in the full-slice setting but encodes less discriminative features from sparse 2D inputs, resulting in poorer few-slice performance. Combining the distillation loss with our aggressive sampling strategy consistently improves results across all scenarios, underscoring the importance of this training scheme for improving robustness and generalization. Additional ablation results are provided in Appendix F.

## 5 Conclusion

We have presented *TetHeart*, a unified framework for 4D cardiac shape and motion reconstruction from 2D CMR slices, operating robustly from full stacks down to a single real-time slice. For the first time, we show that accurate 3D cardiac motion can be inferred from sparse intra-operative CMR data, enabling real-time tracking and guidance during interventions. This has the potential to shift cardiac reconstruction from retrospective analysis to live clinical application, laying the foundation for patient-specific digital twins of the beating heart and paving the way for adaptive, predictive, and personalized image-guided procedures.

As an immediate next step, we plan to deploy *TetHeart* in an interventional MRI suite at our collaborating hospital. Looking ahead, we aim to extend this framework to other dynamic organs, integrate multimodal imaging, and develop predictive digital twins that anticipate motion under both physiological and interventional conditions—advancing toward a new generation of real-time, personalized image-guided medicine.

## Acknowledgment

This work was funded in part by the Swiss National Science Foundation.

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

# A    Datasets and Metrics

## A.1    Unified Dataset

ACDC features CMR sequences of about 30 frames for 100 patients, each covering at least one cardiac cycle and with a slice thickness of 5-10 mm. M&Ms and M&Ms-2 multi-center and multi-vendor datasets containing CMR sequences for 375 and 360 subjects, respectively. While featuring similar resolutions and sequence lengths as ACDC dataset, these two datasets cover a broader range of cardiac pathologies. For these datasets, segmentation masks for the left ventricle (LV), right ventricle (RV), and myocardium (MYO) are provided, but only for the end-diastolic (ED) and end-systolic (ES) frames. As in earlier studies Meng et al. (2023); Yuan et al. (2023); Xiao et al. (2024); Qiao et al. (2025), we register and fit a Statistic Shape Model (SSM) Duan et al. (2019); Bai et al. (2015) to segmentation mask at the ED and ES frames using non-rigid image registration methods to generate cardiac meshes.

## A.2    iCMR Dataset

The iCMR dataset was collected at the Center for Interventional MRI of our collaborating university hospital. The infrastructure used for image acquisition is shown in Fig. 1(c). For each subject, a stack of short-axis (SAX) breath-hold cine slices was acquired at rest on a 1.5T interventional system (Siemens Healthineers, Magnetom Aera) using an accelerated steady-state free precession (SSFP) sequence covering the entire left ventricle. In addition, single SAX cine slices were acquired at mid-ventricular levels with slight angulations relative to the full SAX stack. Because both heart shape and heart rate may vary during intervention, updating the 3D heart model is often required to ensure precise intracardiac manipulations. To mimic heart-rate changes, five volunteers exercised inside the interventional MRI scanner (bending and extending their legs in two parallel plastic gutters for 2–3 minutes), thereby increasing their heart rates by approximately 20–30 beats per minute. Once elevated, single SAX cine slices were reacquired at the same mid-ventricular levels as during rest. For real-time acquisition, the online sequences had lower spatial resolution than the offline short-axis CMR stacks to meet latency constraints.

## A.3    Evaluation Metrics

For evaluation, we compare the predicted 3D mesh obtained by deforming the ED frame to the ES frame with the ground-truth 3D mesh at the ES frame. We use L2-norm Chamfer Distance (CD) to measure 3D reconstruction quality, which is calculated as:

$$CD(P, Q) = \frac{1}{|P|} \sum_{p \in P} \min_{q \in Q} \|p - q\|_2^2 + \frac{1}{|Q|} \sum_{q \in Q} \min_{p \in P} \|q - p\|_2^2.$$

Dice score is also reported to quantify the intra-slice segmentation results, calculated as:

$$DSC(A, B) = \frac{2|A \cap B|}{|A| + |B|}.$$

Besides these quality measures, we also report the Mean-Absolute Error (MAE) of LV/RV ES Volume (LVESV/RVESV), LV/RV ejection fraction (LVEF/RVEF) derived from our predicted meshes. The ejection fraction is calculated as

$$XEF = \frac{XEDV - XESV}{XEDV}, X \in \{LV, RV\},$$

here EDV denotes the volume of the ED meshes. For our method, we use the predicted ED meshes from the reconstruction branch for calculation. For baselines require ground-truth input meshes, we directly use the input meshes for calculation.

The MAE of ES volume and ejection fraction is then calculated as

$$MAE_{esv} = |ESV_p - ESV_{gt}|, \ MAE_{ef} = |EF_p - EF_{gt}|.$$

---

**Algorithm 1:** Adaptive Feature Aggregation (AFA)

---

**Input** : Reference features $F_{2d}^0$, Target features $F_{2d}^t$ (voxels $x \in F_{2d}^0$)
**Input** : Hyperparameters $k_1$ (nearest slices), $k_2$ (nearest positions per slice)
**Output:** Aggregated motion features $V_{\text{motion}}^t$

---

**1 foreach** *voxel $x$ in $F_{2d}^0$* **do**
    `// Step 1:  Identify spatially nearest 2D slices`
**2**    $S \leftarrow$ Select $k_1$ nearest 2D slices from $F_{2d}^t$ relative to $x$ using point-to-plane distance
    `// Step 2:  Retrieve local features within each slice`
**3**    $P \leftarrow \emptyset$
**4**    **foreach** *slice $s \in S$* **do**
**5**        $p_s \leftarrow k_2$ spatially closest positions in slice $s$ to voxel $x$ using point-to-point distance;
        $P \leftarrow P \cup$ Features at $p_s$
**6**    **end**
    `// Step 1&2 together works as the NN-Select(`$F_{2d}^t, k_1, k_2$`) operator`

    `// Step 3:  Define Q, K, V for localized attention`
**7**    $Q \leftarrow F_{2d}^0(x)$                    `// Query is the current voxel feature`
**8**    $K \leftarrow P, V \leftarrow P$              `// Keys and Values from collected set `$P$
    `// Step 4:  Incorporate position embedding and compute MHA`
**9**    $Q, K \leftarrow$ Apply learnable position embedding$(Q, K)$
**10**   $V_{\text{motion}}^t(x) \leftarrow$ `MHAttention`$(Q, K, V)$
**11 end**
**12 return** $V_{motion}^t$

---

The ground-truth volume and ejection fraction are derived from the ground-truth meshes. LVEF and RVEF are representative cardiac function indexes which could be used for clinical diagnosis.

## B   Implementation Details

In all experiments, the slices are resampled by linear interpolation to a spacing of $1.25 \times 1.25$ mm. From this, the AFA module constructs 3D volume with an effective spacing of $1.25 \times 1.25 \times 2$ mm. The AFA module uses a 8-head multi-head attention with learnable 3D position embedding. In the reconstruction branch, a modified nnU-Net Isensee et al. (2021) is used as the image encoder. It comprises five downsampling and upsampling blocks with channel sizes [32, 64, 128, 256, 320]. We use the same GCN structure as in Shen et al. (2021) with 3 layers and 128 channels. The initial tetrahedra resolution is 128. For the motion branch, we use the identical encoding network structure as the reconstruction branch. For the deformation model, the hidden dimension of the GCN and GRU are both set to 128. We use SGD optimizer with an initial learning rate of 0.01, a weight decay of $3e - 5$, and momentum of 0.99. We train all models for 300/150 epochs for the reconstruction/motion learning stage. All experiments are conducted using one V100 GPU.

**AFA Module.** We provide the workflow of how the AFA module is executed in Algorithm 1.

**Two-Stage Weakly Supervised Training.** We provide the two-stage weakly supervised training pipeline in Fig. 7.

## C   Baselines

We compare our method against several existing approaches for 4D shape reconstruction. For the online interventional setting, we adapted several baselines so that they can operate under sparse inputs. **4DMR** Yuan et al. (2023) is an implicit representation method that reconstructs myocardium mesh sequences by optimizing decoupled shape and motion latent codes at test time, minimizing the discrepancy between the predicted mesh and ground-truth contour points. We adapt it by restricting the optimization to incomplete contours extracted from the available few slices, while still using full contours for the initial frame. **MR-Net** Chen

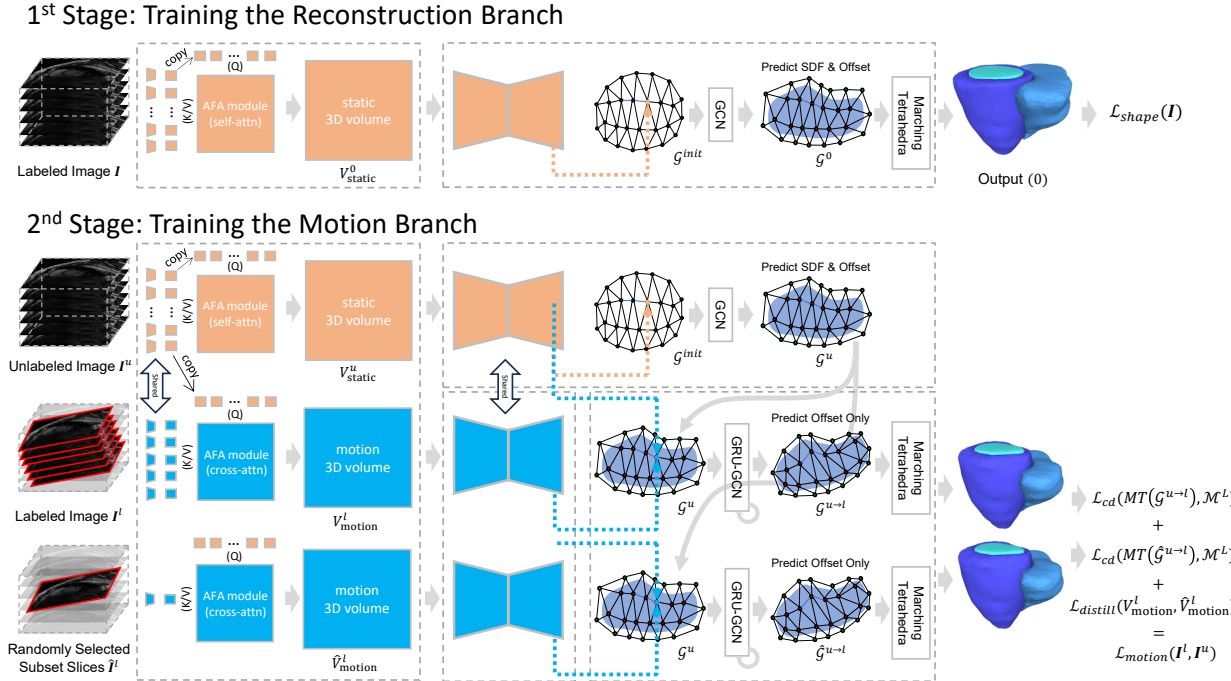

Figure 7: *Two-stage weakly supervised training pipeline.*

et al. (2021) is a template-mesh-based approach that takes ground-truth contour points as input and leverages a hybrid PointNet + 3D CNN architecture to predict vertex deformations. Since MR-Net directly encodes point sets, it can naturally accommodate incomplete inputs, a scenario already considered in the original paper. Although originally designed for static reconstruction, we adapt it for motion estimation by setting the ED-frame mesh as the template. **MulViMotion** Meng et al. (2022b) and **DeepMesh** Meng et al. (2023) are mesh-based approaches that predict deformation fields via 3D CNNs, but only for the myocardium class. They rely on feature concatenation, requiring the initial and target frames to have the same number of slices. We add zero-valued slices to the target frame to enable them to handle few-slice input. Note that this modification only allows them to handle inconsistent slice numbers; it cannot address issues such as scanning-angle discrepancies. For fair comparison, we adopt their SAX-view-only versions as mentioned in the original papers.

In addition, we include two modified baselines derived from shape reconstruction methods: MeshDeform-Net Kong et al. (2021) and DeepCSR Cruz et al. (2021). To adapt them for motion reconstruction, we integrate into both the same nnU-Net backbone, AFA module, deformation model, and training pipeline as in our method. Since their main differences from our full framework lie in the shape representation, we denote them as **Ours-Mesh** and **Ours-SDF**.

For the traditional offline setting, in addition to the above methods, we further compare against B-spline Free-Form Deformation (FFD) Rueckert et al. (2002), diffeomorphic Demons (dDemons) Vercauteren et al. (2007), MR-Net w. nnU-Net, MulViMotion, and DeepMesh. **FFD** and **dDemons** are classical registration algorithms that have been widely used in recent cardiac motion tracking studies Bai et al. (2020); Qin et al. (2020); Puyol-Antón et al. (2018). **MR-Net w. nnU-Net** adopts the MR-Net architecture but replaces ground-truth contours with those predicted by nnU-Net, forming a two-stage automated pipeline.

Unless explicitly stated otherwise, all methods are jointly trained on the unified dataset.

# D  Complementary Experiment Results

**Disease Definition.** A normal heart should satisfy the following criteria: LV EF > 50% and RV EF > 40%. A subject with dilated cardiomyopathy should have LV EF < 40%. This is not a strict definition; a more

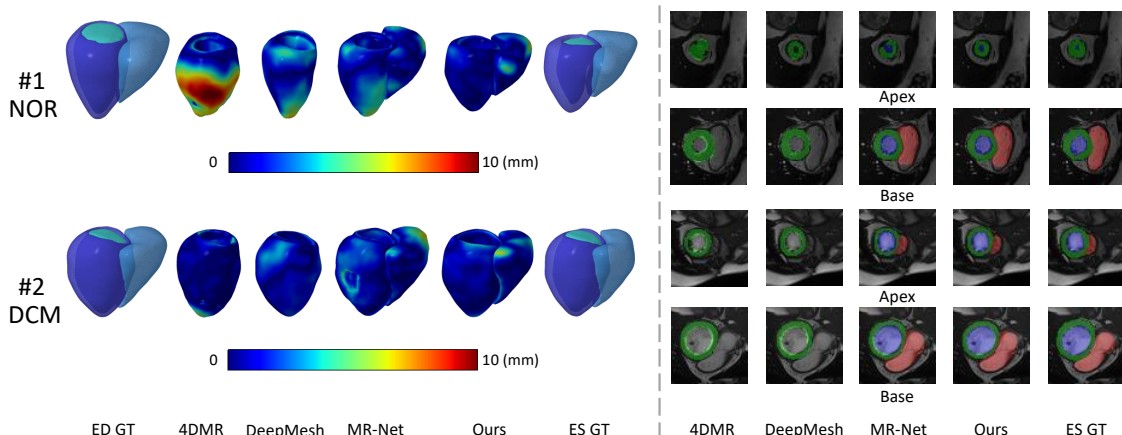

Figure 8: Full-Slice Comparison. (Left) We predict 3D meshes by deforming from ED to ES frame. Color indicates the magnitude of point-to-surface error in mm. (Right) Segmentation results on slices at the apex and base location. Note 4DMR and DeepMesh can only output for myocardium.

| Method | Training Dataset | | | | Myocardium CD ($mm^2$) ↓ | | |
| --- | --- | --- | --- | --- | --- | --- | --- |
| | ACDC | M&Ms | M&Ms-2 | 4DM | 1-Slice | 5-Slice | Full-Slice |
| FFD | - | - | - | - | - | - | 11.51 |
| dDemons | - | - | - | - | - | - | 12.46 |
| 4DMR | ✓ | ✓ | ✓ | - | 24.00 | 19.03 | 13.97 |
| MR-Net | ✓ | ✓ | ✓ | - | 15.87 | 11.56 | 7.97 |
| DeepMesh | ✓ | ✓ | ✓ | - | - | - | 8.32 |
| Ours-Mesh | ✓ | ✓ | ✓ | - | 13.31 | 10.32 | 5.97 |
| Ours-SDF | ✓ | ✓ | ✓ | - | 13.78 | 10.78 | 6.24 |
| | ✓ | - | - | - | 25.43 | 19.42 | 13.81 |
| Ours | - | ✓ | - | - | 16.91 | 12.54 | 8.23 |
| | - | - | ✓ | - | 17.88 | 13.03 | 8.83 |
| | ✓ | ✓ | ✓ | - | **11.33** | **9.08** | **5.51** |
| Ours (Oracle) | - | - | - | ✓ | 6.24 | 4.87 | 2.70 |

Table 6: *Zero-shot evaluation results on public 4DM dataset.* Check marks denote the dataset used for training. Oracle is trained and evaluated on 4DM.

rigorous classification criterion would require a comprehensive analysis of the ratio between volume and body surface area, along with other indicators. However, verifying whether the predicted ejection fraction from *TetHeart* satisfies this approximate criterion can further demonstrate the practicality of our model.

As can be seen from Fig. 4, even with a single slice, *TetHeart* can predict cardiac functional indices that satisfy these criteria, highlighting the practicality of our model.

**Visualization Results Using Full Slices.** In Fig. 8, we show qualitative results on the same hearts as in Fig. 3, but using full slices instead of a single slice.

**Statistical Significance Evaluation.** We perform a paired t-test to evaluate the statistical significance of our method.

- **Tab. 1, motion reconstruction, 1-slice setting**. For the 1-slice setting of Tab. 1, our method achieves statistically significant improvements ($p < 0.05$) when comparing the average Chamfer Distance to all competitors. Specifically, 4DMR, MR-Net, MulViMotion, and DeepMesh have $p$-values smaller than 0.001, while Ours-Mesh ($p = 0.0347$) and Ours-SDF ($p = 0.0213$).

- **Tab. 3, motion reconstruction, full-slice setting.** For the full-slice setting of Tab. 3, when comparing the average Chamfer Distance, the $p$-values for all competitors are <0.001 (FFD, dDemons, 4DMR, MR-Net, MulViMotion, DeepMesh), 0.2331 (Ours-Mesh), 0.1988 (Ours-SDF). As expected, when the image information is relatively sufficient, the variants of our method with different representations perform relatively close, and the differences are not statistically significant.

| Method | 4DMR | MR-Net | Ours-Mesh | Ours-SDF | Ours |
|---|---|---|---|---|---|
| GLS MAE (1-Slice, %) | 4.47 | 3.73 | 2.65 | 3.08 | **2.18** |

Table 7: *Strain analysis results on M&Ms dataset.*

- **Tab. 4, iCMR external evaluation.** For the external evaluation results in Tab. 4, when comparing the average Chamfer Distance, the *p*-values for all competitors are <0.001 (4DMR, MR-Net, Ours trained with different subset of training dataset), 0.0192 (Ours-Mesh), 0.0087 (Ours-SDF).

- **Tab. 5 (right), ablation study.** For the ablation studies, under the 1-slice setting, the *p*-values of each ablation variants are 0.0172 (-pe), <0.001 (-image feature/-shared encoding), 0.1217 (-distillation loss). Under the full-slice, the *p*-values are 0.4123 (-AFA), 0.2918 (-pe), <0.001 (-image feature/-shared encoding). The value is not meaningful for the distillation variant as its performance is better than Ours.

**Strain Analysis.** Figure 3 and 8 justify the plausibility of cardiac motion by measuring volumetric changes and the myocardial volume curves partially reflect quasi-incompressibility. We further verify the accuracy of the estimated motion using ED-to-ES strain analysis. The Mean Absolute Error of Global Longitudinal Strain in reported in Tab. 7. As could be seen, ours performs best. This further validates the effectiveness of our method.

# E    External Evaluation Results

**4DM Dataset.** As 4DM provides full-sequence annotations, but only for the myocardium, we evaluate full-sequence prediction performance for the myocardium subclass rather than focusing solely on the ES frame, and report the results in Tab. 6. Here, check marks denote the dataset used for training. *Oracle* refers to training and testing directly on 4DM, serving as an upper bound for performance.

As shown in the table, our model demonstrates the best generalization ability, achieving top performance in both few-slice and full-slice settings. When compared to the best competitor MR-Net, our model generalizes significantly better and achieves a 30% improvement. In contrast, 4DMR exhibits poor generalization, likely due to significant distribution shifts across datasets and the limited representational capacity of its simple MLP architecture. While MR-Net and DeepMesh show some generalization ability, they still perform worse than ours.

Consistent with findings from previous evaluations, training with a larger and more diverse dataset improves generalization. Moreover, representing the heart as tetrahedra enables more effective spatial information retention and exchange, particularly in scenarios with limited slice input.

# F    Extended Ablation Studies

**The Impact of AFA Configuration.** In Tab. 8, we show the impact of the number of slices in the AFA module and the number of selected positions per slice on both performance and speed (measured on an NVIDIA V100 GPU). As shown in the table, our default configuration achieves a good balance between accuracy and efficiency. Further increasing the number of slices or selected positions brings only limited performance gains but leads to a slowdown. One point worth noting is that, to implement our AFA module, we use a relatively simple strategy of sorting and selecting the top-$k$ elements. Therefore, for $|S| = 1, |G| = 5^2$ and $|S| = 3, |G| = 3^2$, although the two configurations use a similar number of features, the latter runs slower because it performs sorting multiple times. More advanced strategies or precomputation could potentially narrow this gap and enable further acceleration.

**The Influence of the Input Slice Position.** In the main text, for ease of comparison and to ensure fairness, we use central slices as model inputs to evaluate the performance of different methods. Here we investigate how the input slice position affects the accuracy of motion inference. The results are shown in Tab. 9. The numbers indicate the offset of the input slice relative to the central slice: $-$ denotes a shift

| Model | $|S|$ | $|G|$ | Myocardium CD & FPS ($mm^2 \downarrow$ & $img/s \uparrow$) | | | | | |
| | | | 1-Slice | | 5-Slice | | Full-Slice | |
|---|---|---|---|---|---|---|---|---|
| *TetHeart* | 1 | $1^2$ | 10.34 | 15.1 | 8.45 | 15.1 | 4.56 | 15.1 |
| | 1 | $3^2$ | 9.98 | 15.0 | 8.22 | 14.9 | 4.43 | 14.9 |
| | 1 | $5^2$ | 9.93 | 14.8 | 8.13 | 14.7 | 4.39 | 14.6 |
| | 3 | $1^2$ | 10.32 | 15.0 | 8.19 | 13.0 | 4.37 | 13.0 |
| | 3* | $3^{2}$* | 9.76 | 14.9 | 7.93 | 12.1 | 4.11 | 12.0 |
| | 3 | $5^2$ | 9.73 | 14.7 | 7.86 | 10.8 | 4.05 | 10.7 |
| | 5 | $1^2$ | 10.27 | 15.0 | 8.17 | 11.9 | 4.33 | 11.8 |
| | 5 | $3^2$ | 9.74 | 14.8 | 7.81 | 10.4 | 4.09 | 10.4 |
| | 5 | $5^2$ | 9.72 | 14.7 | 7.83 | 8.7 | 4.10 | 8.7 |
| | 7 | $5^2$ | 9.76 | 14.6 | 7.85 | 8.6 | 4.07 | 7.3 |

Table 8: *Impact of different AFA configuration evaluated on the M&Ms dataset.* $|S|$: the number of slice. $|G|$: the number of positions selected from each slice. *: default setting.

toward the apex, and + denotes a shift toward the base. For example, suppose we have total 11 slices, then the central slice is the 6-th slice, the +1 slice is the 7-th slice, the −1 slice is the 5-th slice.

As seen from the table, although the model still achieves good motion inference performance, its accuracy steadily decreases as the offset increases. These experimental results suggest that slices at different positions contain different amounts of motion information, which may be more focused on local regions. If only single slice is available, selecting it close to the central position yields better overall motion inference. This conclusion provides guidance on choosing suitable scan positions in real-world intervention workflows. For multiple slices, we could conduct similar experiments to explore which slice combinations are most effective. However, due to the large number of possible combinations, we provide only one general conclusion here: using an appropriate slice sampling interval can improve model performance compared with using adjacent slices.

| Slice Position | -3 | -2 | -1 | Central | +1 | +2 | +3 |
|---|---|---|---|---|---|---|---|
| Myo CD ($mm^2 \downarrow$) | 15.76 | 13.62 | 10.01 | 9.76 | 10.29 | 10.81 | 12.22 |

Table 9: *Impact of input slice position evaluated on the M&Ms dataset.* The numbers indicate the offset of the input slice relative to the central slice. − denotes a shift towards the apex, and + denotes a shift towards the base.

**Distillation Loss.** We provide the results of full-stack-only training in Tab. 10. As expected, full-stack-only training performed best on the full-slice setting, but the difference from our complete model was not significant. The paired t-test (Full-stack-only v.s. Ours) yields a *p*-value of 0.1563. This result indicates that, given the same architecture, our full model which leverages a two-stage weakly supervised training scheme combined with the distillation loss, achieves robust performance in the challenging few-slice setting at only a marginal cost to full-slice performance.

| Method | M&Ms Myo CD ($mm^2$) $\downarrow$ | | |
| | 1-Slice | 5-Slice | Full-Slice |
|---|---|---|---|
| Ours | 9.76 | 7.93 | 4.11 |
| - distillation loss | 10.34 | 8.12 | 4.03 |
| Full-stack-only training | - | - | 3.94 |

Table 10: *Distillation Loss Effects.*

**Weight sharing strategy.** In Tab. 11, we provide a more detailed ablation about the weight sharing strategy. In fact, we can also initialize the motion branch from the parameters of the reconstruction branch without sharing the weights and optimize the two branches separately. However, this approach did not help. Therefore, we adopted the design of shared weight, reducing the number of parameters while having the same performance.

| Method | M&Ms Myo CD (mm$^2$) ↓ | | |
| --- | --- | --- | --- |
| | 1-Slice | 5-Slice | Full-Slice |
| Ours | 9.76 | 7.93 | 4.11 |
| - shared encoding network (init motion from reconstruction) | 9.64 | 7.87 | 4.07 |
| - shared encoding network (random init) | 20.66 | 11.63 | 6.43 |

Table 11: *Detailed weight sharing ablation.*

Although static reconstruction and motion estimation are conceptually different tasks, they are both concerned with predicting vertex-level geometric properties in a common 3D deformable representation space. In our framework based on deformable tetrahedral meshes, this connection becomes particularly explicit: Static reconstruction predicts vertex offsets and SDF-related attributes on the template tetrahedra $\mathcal{G}^{init}$, while motion estimation predicts vertex displacements of the tetrahedra $\mathcal{G}^0$. In both cases, the underlying objective is to infer meaningful vertex-wise geometric transformations conditioned on image observations.

From this perspective, the two branches can be interpreted as learning two stages of the same geometric inference process. Empirically, this shared representation helps transfer priors learned from the reconstruction task to the more challenging motion estimation task. Another important aspect of our design is that parameter sharing is limited to the feature extraction backbone, while the task-specific reasoning modules are explicitly decoupled: The static branch uses a GCN-based decoder, whereas the motion branch employs a GRU-GCN structure. This separation preserves task-specific modeling capacity.

Overall, we find that shared weights provide a simple yet effective way to couple two closely related geometric prediction tasks, improving parameter efficiency without sacrificing expressiveness.

**Iterative Motion Refinement.** As described in Sec. 3.4, the vertex update process (Eqs. 3–6) is performed iteratively twice. This is because a single update step is insufficient to accurately handle relatively large deformations, while iterative refinement allows the initial shape to progressively approach the target shape. We provide in Tab. 12 an analysis of the effect of the number of update iterations on performance below on the M&Ms dataset.

| Method | 1-slice CD ($mm^2$) ↓ | | | Full-slice CD ($mm^2$) ↓ | | |
| --- | --- | --- | --- | --- | --- | --- |
| | Myo | LV | RV | Myo | LV | RV |
| Ours (default, repeat=2) | 9.76 | 12.37 | 21.42 | 4.11 | 4.85 | 13.68 |
| repeat=1 | 11.36 | 14.31 | 24.26 | 4.91 | 5.37 | 15.23 |
| repeat=3 | 9.70 | 12.49 | 21.32 | 4.17 | 4.93 | 13.59 |

Table 12: *The impact of different repeat counts on the M&Ms dataset.*

This confirms that using only one update step leads to a noticeable performance drop, whereas increasing the number of iterations beyond two does not provide further improvement. One possible explanation is that cardiac motion itself is physiologically constrained, such that the magnitude of deformation between frames is naturally limited.

At the same time, we acknowledge that this design still has limitations. The hyperparameters were chosen given the characteristics of the datasets we currently have. Therefore, when encountering out-of-distribution data, its performance may decline. Meanwhile, if one wants to model other dynamic objects, the appropriate hyperparameters may be different. How to propose a more robust motion model is an interesting study. But for the task at hand, our design already delivers good performance.

**Category-Wise Ablation Results.** In Tab. 13, we present more detailed results on the unified dataset as a supplement to Tab. 5 in the main text. As shown in the table, across all datasets and all categories, the conclusions remain consistent with those in the main text: the design of the AFA module, parameter sharing, and the use of the distillation loss all contribute to improving the final performance of our model.

**Relax the initial full stack requirement.** Though in a standard clinical intervention workflow, performing a full-stack cardiac scan before the procedure is a standard practice. Nevertheless, investigating whether

| Method | Myocardium CD (mm$^2$) ↓ | | | LV CD (mm$^2$) ↓ | | | RV CD (mm$^2$) ↓ | | |
|---|---|---|---|---|---|---|---|---|---|
| | 1-Slice | 5-Slice | Full-Slice | 1-Slice | 5-Slice | Full-Slice | 1-Slice | 5-Slice | Full-Slice |
| ACDC | | | | | | | | | |
| Ours | 15.24 | 10.22 | 6.63 | 23.65 | 18.31 | 11.51 | 40.64 | 32.61 | 22.15 |
| - AFA module | - | - | 6.55 | - | - | 11.58 | - | - | 22.30 |
| - position embedding | 16.60 | 11.33 | 6.84 | 25.32 | 20.14 | 12.73 | 42.23 | 34.19 | 23.59 |
| - image feature in query $Q$ | 24.10 | 15.96 | 9.39 | 36.76 | 28.39 | 17.44 | 58.31 | 43.21 | 32.32 |
| - shared encoding network | 32.23 | 15.24 | 10.01 | 49.12 | 28.99 | 18.59 | 75.92 | 44.81 | 34.44 |
| - distillation loss | 16.13 | 10.48 | 6.49 | 25.07 | 18.78 | 11.28 | 43.08 | 33.43 | 21.71 |
| M&Ms | | | | | | | | | |
| Ours | 9.76 | 7.93 | 4.11 | 12.37 | 9.72 | 4.85 | 21.42 | 17.39 | 13.68 |
| - AFA module | - | - | 4.22 | - | - | 4.79 | - | - | 13.74 |
| - position embedding | 11.23 | 9.04 | 4.67 | 13.35 | 10.34 | 5.67 | 22.72 | 18.84 | 14.93 |
| - image feature in query $Q$ | 16.67 | 12.37 | 6.44 | 19.76 | 14.17 | 7.82 | 33.63 | 23.81 | 20.60 |
| - shared encoding network | 20.66 | 11.63 | 6.43 | 24.56 | 14.24 | 7.77 | 41.80 | 24.12 | 20.45 |
| - distillation loss | 10.34 | 8.12 | 4.03 | 13.11 | 9.94 | 4.75 | 22.70 | 17.79 | 13.41 |
| M&Ms-2 | | | | | | | | | |
| Ours | 10.12 | 7.30 | 4.02 | 13.89 | 9.96 | 5.04 | 20.27 | 15.58 | 10.82 |
| - AFA module | - | - | 4.17 | - | - | 5.07 | - | - | 10.76 |
| - position embedding | 11.93 | 8.41 | 4.74 | 14.98 | 10.40 | 5.69 | 21.91 | 16.72 | 11.65 |
| - image feature in query $Q$ | 17.34 | 11.38 | 6.31 | 22.32 | 13.57 | 7.57 | 32.65 | 21.40 | 15.49 |
| - shared encoding network | 21.49 | 11.09 | 6.54 | 26.96 | 13.64 | 7.85 | 39.44 | 20.73 | 16.07 |
| - distillation loss | 10.95 | 7.47 | 4.11 | 15.01 | 10.19 | 5.05 | 21.89 | 15.94 | 11.06 |

Table 13: *More comprehensive ablation study results on the unified dataset.*

it is possible to relax or even eliminate the requirement for the initial full CMR stack is an interesting problem. Conceptually, the model could instead start from a generic template tetrahedral mesh and directly update the vertex positions based on the observed few-slice inputs, thereby removing the dependency on patient-specific full-stack initialization.

Thus, we conducted a preliminary experiments to investigate this intriguing possibility. As our current framework was not designed with this specifc setting in mind, we modified it in two ways: (1) replacing the image-feature-based queries in the AFA module with purely spatial positional queries, and (2) removing the static–dynamic feature concatenation in Eq. (3), making the deformation prediction rely on only motion features. As in Table 1 of the paper, we compared the predicted 3D meshes by deforming from the initial state to ES frame to the ground-truth 3D meshes, and present the results in Tab. 14. The initial state is the ED reconstruction in the default setting, and a generic template tetrahedral mesh after coarse registration to the segmentation in the modified variant.

| Method | Myo CD ($mm^2$) | LV ($mm^2$) | RV ($mm^2$) |
|---|---|---|---|
| Ours | 9.76 | 12.37 | 21.42 |
| from template | 16.42 | 18.73 | 34.61 |

Table 14: *A probe experiment about whether it is possible to relax the initial full stack requirement.* from template: start from a coarsely registered template tetrahedral mesh.

Although we can still achieve a marginally satisfactory performance, there is a clear performance drop compared to our default setting. The reason is straightforward: Due to the large variability in cardiac anatomy across patients, a template mesh cannot effectively capture all patient-specific features. Moreover, the absence of image features also prevents the model from using rich image correlation information to improve performance. In contrast, the mesh reconstructed from the ED full-stack data is able to better capture this information and thus provide a stronger morphological prior, and the features of the initial image have also been fully utilized through the AFA module. Consequently, when only limited information is available in the subsequent frames, the latter demonstrates superior performance.

**Rationale Behind the Training Strategy.** In Sec. 3.5, when training the motion branch, we propose an aggressive augmentation strategy by using slice subsampling and random downsampling/rotation/noise jittering to mimic the interventional scenario. The rationale behind this strategy is given in Fig. 9. As can be seen, in terms of image quality, the interventional slice samples in the iCMR dataset differ from standard SAX images mainly in resolution and sharpness. Thus, our proposed training strategy can effectively bridge the domain gap between conventional SAX slices and interventional slices. This is also validated by the ablation results in Tab. 5(left), where the proposed training strategy effectively handles the simulated interventional artifacts.

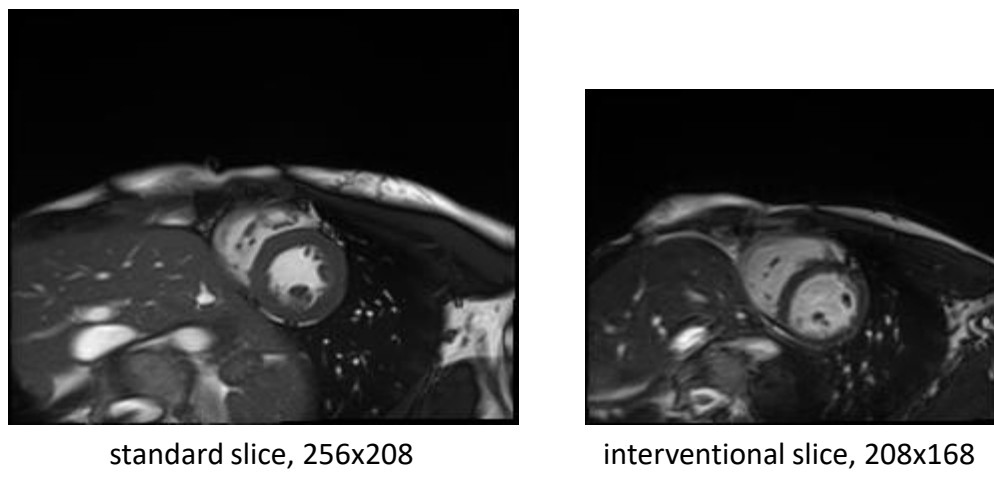

standard slice, 256x208          interventional slice, 208x168

Figure 9: Normal SAX and interventional slice comparison.

**A Not-so-Successful Case.** In Fig. 10, we present a less successful reconstruction example of a Myocardial Infarction case. As observed from the images, when given only a single slice as input, our model maintains decent accuracy in predicting the ejection fraction, but falls short in accurately estimating the absolute volume. Incorporating disease-specific prior knowledge or developing a more advanced temporal module could potentially alleviate this issue.

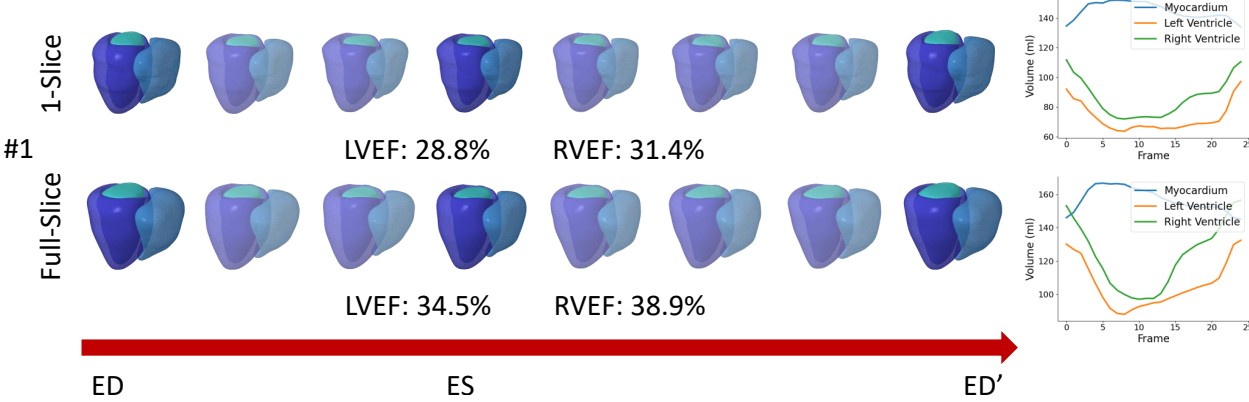

Figure 10: *A not-so-successful motion sequence prediction example.*

# G   More Visualization Results

We provide video results in the 'video' subfolder. 'single_and_full_slice_comparison.mp4' presents video results for the NOR and DCM samples used in Fig. 4. 'icmr_results.mp4' presents video results for the samples from iCMR samples used in Fig. 6, which uses a single slice with slight angulations relative to

the full SAX stack for motion inference. We also provide the video results for more diseases, including Myocardial Infarction (MINF), Abnormal Right Ventricle (ARV), in folder 'video/more_sequences'. Each subject is named 'XX_sYY.mp4', where XX is the disease category and YY is the number of slices used to recover motion.

## H    Limitation

Our method achieves promising results in both full-stack and few-slice settings, surpassing previous methods. However, by examining our own model, we observe that although we have proposed several techniques, there still remains a performance gap between motion reconstruction using only a few slices and using the full stack. We plan to introduce physical constraints as prior knowledge into the model to reduce this gap. Another option is to use a generative model to progressively generate the mesh sequence, complementing our current feed-forward reconstruction process. However, we need to strike a good balance between reconstruction speed and quality if we aim for real-time applications.

While the distillation training strategy can improve performance, it also carries the risk of introducing population-level statistics that may lead to hallucinated anatomy, producing 3D cardiac shapes with averaged features. Such a tendency might obscure unique pathological anomalies. Our framework's flexibility in handling arbitrary inputs provides a possible mitigation: dynamically adjusting scan planes to capture missing details during intervention. Although effective integration of this temporal information requires a dedicated temporal module, we view this as a valuable future extension. We leave these as future work.

## I    Broader Impact

On the positive side, our method may facilitate real-time, image-guided cardiac interventions by improving imaging efficiency and reducing acquisition requirements, which could potentially enhance procedural safety and accessibility. However, we also acknowledge that model failures or unreliable predictions could lead to incorrect clinical interpretation or procedural guidance. Such risks may become more pronounced in out-of-distribution cases, rare anatomical variations, low-quality acquisitions, or scenarios that are underrepresented in the training data.

To mitigate these concerns, we emphasize that, given our method's current state, it is intended to function as an assistive tool rather than a fully autonomous clinical system. From the model part, as our model is flexible to take arbitrary slices as input, we can dynamically adjust scan planes to capture missing details during intervention to reduce error. Although the integration of temporal information requires us to develop a temporal module. However, no matter how the model is improved, errors can never be completely avoided. Thus in practical deployment, model outputs should always be reviewed by qualified clinicians and integrated with other clinical observations. Potential safeguards, including uncertainty estimation, confidence-aware quality control, and additional validation across diverse patient populations and imaging devices can also be introduced before clinical adoption.

