# OpenReview forum: "End-to-End 4D Heart Mesh Recovery Across Full-Stack and Sparse Cardiac MRI"
_TMLR — Accepted by TMLR_

### Review · Reviewer_kYn2 · 2026-04-08

**Summary Of Contributions:**

This paper introduces TetHeart, the first unified end-to-end framework for 4D cardiac mesh reconstruction from both full-stack CMR (offline) and sparse-slice real-time MRI (intra-procedural). The key contributions are: (1) an Attentive 2D-3D Feature Assembler (AFA) that aggregates information from an arbitrary number of 2D slices at arbitrary positions; (2) a distillation strategy that transfers knowledge from dense to sparse inputs; (3) a weakly supervised motion learning scheme that requires annotations only at key cardiac phases (ED/ES). Extensive validation demonstrates state-of-the-art performance in both full-slice and few-slice settings.

**Audience:**

Yes

**Audience Explanation:**

The paper would likely attract interest from a subset of the TMLR audience, particularly those working in medical imaging, geometric deep learning, and spatiotemporal modeling. The problem of reconstructing dynamic 3D structures from sparse observations is broadly relevant and extends beyond cardiac imaging to other domains involving partial observations and real-time inference.

**Broader Impact Concerns:**

The proposed method positively enables real-time, image-guided cardiac interventions, potentially improving clinical outcomes and procedural safety. Nevertheless, concerns remain regarding its reliability and deployment in high-stakes medical environments, where model errors could lead to serious consequences. It would be better if the authors voluntarily provide an impact statement that addresses potential issues and mitigation strategies in real-world deployment and usage.

**Claims And Evidence:**

Yes

**Claims Explanation:**

1. This paper addresses a clinically critical and technically underexplored problem: reconstructing 3D cardiac motion from sparse, real-time MRI slices during interventions. This paper proposes TetHeart, which incorporates the AFA module, a distillation strategy, and a weakly supervised motion learning scheme.
2. This paper is well-organized, featuring clear and informative figures, detailed method sections, algorithm pseudocode, and thorough appendices that cover datasets, implementation, and baselines.
3. The experimental evaluation is thorough, including multiple public datasets, external validation without retraining, ablation studies, and clinically meaningful metrics, which collectively strengthen the empirical credibility of the work.

**Requested Changes:**

1.	There are several typos and grammatical errors in the paper. I have listed some of them here. Please carefully check the entire manuscript and correct them.
Page 1: “a limitation we observed in practice” -> “which is a limitation we observed in practice”;
Page 3 and 4: “at sufficient frame rate” -> “at a sufficient frame rate”;
Page 3: “due to it can capture dynamic processes” -> “due to its ability to capture dynamic processes”;
Page 4: “at lower spatial resolution” -> “at a lower spatial resolution”;
Page 7: “despite these challenges as will be shown in” -> “despite these challenges, as will be shown in”.
2.	Figure 2 illustrates the architecture of the proposed TetHeart. It clearly shows the static and dynamic branches, but it lacks a demonstration of the distillation loss and two-stage weakly supervised training. Please modify the figure 2 or add a new figure to illustrate the training and inference state of the model, in order to fully visualize the proposed innovations.
3.	In Figure 6, the caption and legend fonts in the volume–time curves are too small, and the image resolution is insufficient.
4.	Why are only these three scenarios (1-slice, 5-slice, and full-slice) considered? Furthermore, how are the slice positions selected for the 1-slice and 5-slice cases?

---

> ### Author Response · Authors · 2026-05-28
> **Rebuttal**
>
> We appreciate the reviewer’s positive assessment of our work and their recognition of the clinical importance and technical relevance of the problem we address. We are encouraged that the reviewer acknowledged the effectiveness of TetHeart and its key components. We would like to address their concerns below.
>
> > **Q1**: Typos.
>
> **A1**: Thanks for pointing them out. We have revised the manuscript to fix the typos.
>
> > **Q2**: Figures.
>
> **A2**: Thanks for your suggestion, we have added a new figure (Figure 7) in the appendix to supplement Figure 2 to better show the training state of the model. The resolution of Figure 6 is also increased for better visibility.
>
> > **Q3**: Slice selection strategy.
>
> **A3**: Our primary focus is on the two extreme settings: 1-slice and full-slice. The 1-slice setting corresponds to the scenario most often used in clinical practice, while the full-slice setting represents the conventional experimental setup. However, as acquisition technology continues to improve, the number of slices that can be simultaneously obtained is likely to gradually increase. Therefore, exploring intermediate slice-count settings provides a useful indication of how the model will scale and perform under future acquisition conditions. The reason for selecting the 5-slice setting is that most samples in our dataset contain around 10 slices. We therefore chose a simple intermediate value for detailed evaluation, while Fig. 5 illustrates the overall performance trend as the number of slices varies.
>
> Regarding slice position selection, Section 4.2 states that we use the “central 1 or 5 slices.” Specifically, assuming a total of 11 slices indexed from 1, the central 1 slice corresponds to slice 6, while the central 5 slices correspond to slices 4-8. This choice was mainly made to ensure consistency in evaluation. In practice, as also mentioned in Section 4.2 and validated in Section 4.4, “slices at arbitrary positions could have been used.” To further clarify this point, we have added a more detailed ablation study on slice positions in Appendix F. These additional results help better characterize both the strengths and limitations of our model, while also providing insights into slice position selection for future applications.
>
> > **Q4**: Broader impact concern.
>
> **A4**: This is an important consideration. Thus, we have added a discussion on the broader impact of our method in the final section of the appendix. We discuss both the potential clinical benefits and the limitations associated with real-world deployment in high-stakes medical environments. We believe these discussions provide a balanced perspective on both the opportunities and the limitations of our approach, while clarifying the precautions necessary for safe real-world usage.

---

### Review · Reviewer_xTXX · 2026-05-04

**Summary Of Contributions:**

In this paper, the authors propose TetHeart, an end-to-end framework for reconstructing 4D cardiac motion from both full-stack cardiac MRI and sparse 2D slice observations. Existing approaches generally require dense volumetric inputs and are therefore unsuitable for real-time interventional settings where only a small number of slices can be acquired. To address the challenge, TetHeart introduces three main components: (1) an Attentive 2D–3D Feature Assembler (AFA) for aggregating features from arbitrarily positioned slices; (2) a full-to-sparse distillation strategy to improve reconstruction under extreme sparsity; and (3) a weakly supervised motion learning scheme requiring annotations only at key cardiac phases. Experiments on multiple datasets demonstrate strong performance in both full-stack and sparse-slice settings.

**Audience:**

Yes

**Audience Explanation:**

The paper addresses an important problem at the intersection of machine learning and medical imaging, particularly in clinically relevant settings such as real-time cardiac interventions. This work is highly relevant to researchers in computer vision, medical image analysis, and learning from imperfect data.

**Claims And Evidence:**

Yes

**Claims Explanation:**

The paper addresses a clinically important and underexplored problem: reconstructing full 4D cardiac motion from sparse MRI observations in interventional settings. It clearly highlights the limitation of existing methods that require full CMR stacks and proposes a solution that directly tackles this challenge. The paper provides strong and convincing empirical evidence, showing consistent improvements across multiple public datasets and external datasets without retraining. The evaluation includes both geometric and clinically relevant metrics, further strengthening the results.

**Requested Changes:**

- The proposed method relies on an initial full CMR stack to construct the patient-specific mesh. Is it possible to relax or eliminate this requirement for the initial frame?

- The proposed Attentive 2D–3D Feature Assembler (AFA) appears to rely heavily on standard multi-head attention, with modifications such as restricting attention to local neighbors across slices and spatial positions. While these design choices are reasonable and improve efficiency, they are conceptually similar to existing attention mechanisms. Thus, the technical novelty of AFA seems limited.

- The motion model is based on deforming an initial mesh over time. It is unclear how well this approach handles longer temporal lengths with large deformations.

- The current full-to-sparse distillation loss formulation enforces similarity in feature space, but it is not clear whether this is the most appropriate supervision signal for motion reconstruction. It would be interesting to explore whether directly distilling geometric or motion outputs (rather than intermediate features) leads to better results.

---

> ### Author Response · Authors · 2026-05-28
> **Rebuttal 1**
>
> We appreciate the reviewer’s positive assessment of our work and their confirmation of the clinical importance and relevance of the problem we address. We also thank the reviewer for highlighting the strong empirical validation across multiple public and external datasets. We would like to address the concerns below.
>
> > **Q1**: Relax the initial full stack requirement.
>
> **A1**: This is an interesting question even though, in a standard clinical intervention workflow, performing a full-stack cardiac scan before the procedure is a standard practice. Therefore, assuming the availability of a complete stack does not detract from the usefulness of the the technique we propose.
>
> Nevertheless, in the short time available to us, we attempted to answer the reviewer's question because, in principle, relaxing or even eliminating the requirement for the initial full CMR stack should be possible. Conceptually, the model could instead start from a generic template tetrahedral mesh and directly update the vertex positions based on the observed few-slice inputs, thereby removing the dependency on patient-specific full-stack initialization.
>
> Thus, we conducted a preliminary experiments to investigate this intriguing possibility. As our current framework was not designed with this specifc setting in mind, we modified it in two ways: (1) replacing the image-feature-based queries in the AFA module with purely spatial positional queries, and (2) removing the static–dynamic feature concatenation in Eq. (3), making the deformation prediction rely on only motion features.
>
> The results are shown in the table below and also added to Appendix F. As in Table 1 of the paper, we compared the predicted 3D meshes by deforming from the initial state to ES frame to the ground-truth 3D meshes. The initial state is the ED reconstruction in the default setting, and a generic template tetrahedral mesh after coarse registration to the segmentation in the modified variant.
>
> |     | Myo CD $\downarrow$ | LV CD $\downarrow$ | RV CD $\downarrow$ |
> | --- | :---: | :---: | :---: |
> | default | 9.76 | 12.37 | 21.42 |
> | from template | 16.42 | 18.73 | 34.61 |
>
> Although we can still achieve a marginally satisfactory performance, there is a clear performance drop compared to our default setting. The reason is straightforward: Due to the large variability in cardiac anatomy across patients, a template mesh cannot effectively capture all patient-specific features. Moreover, the absence of image features also prevents the model from using rich image correlation information to improve performance. In contrast, the mesh reconstructed from the ED full-stack data is able to better capture this information and thus provide a stronger morphological prior, and the features of the initial image have also been fully utilized through the AFA module. Consequently, when only limited information is available in the subsequent frames, the latter demonstrates superior performance.
>
> > **Q2**: Technical novelty of Attentive 2D–3D Feature Assembler.
>
> **A2**: We acknowledge this and have explicitly cited in Section 3.2 of the manuscript that the design of the AFA module is largely inspired by Swin Transformer. Therefore, similarities at the technical and implementation level with prior attention mechanisms are natural. However, we do not see this as major issue because the main novelty does not lie in proposing an entirely new attention operator from scratch, but rather in designing a task-specific feature integration mechanism for a previously underexplored problem setting. This is not different from many modern vision methods in which the core building blocks (e.g., convolutions, self-attention, MLPs) are often shared across different approaches. What is truly critical is how these components are adapted and structured to address the unique challenges of a specific target problem.
>
> In our setting, the key challenge was to effectively aggregate 3D information from sparse 2D slice features with variable slice numbers and positions, while maintaining computational efficiency suitable for practical usage. The proposed AFA module was specifically designed around these requirements by restricting attention interactions to relevant local neighborhoods. This design enables efficient 2D–3D feature assembly. Therefore, although AFA shares technical similarities with existing attention mechanisms, we believe the novelty lies in identifying and formulating this clinically relevant sparse-slice-to-3D reconstruction problem and introducing a targeted attention-based solution tailored to the structural characteristics of the task.

---

> > ### Author Response · Authors · 2026-05-28
> > **Rebuttal 2**
> >
> > > **Q3**: Handling longer temporal lengths with large deformations.
> >
> > **A3**: Our model's performance does not depend on the temporal sequence length, since the framework only requires the initial frame and the current one as inputs. Therefore, increasing the temporal length does not by itself directly increase the prediction difficulty. What is critical is that we can recover the heart deformation at any point on the cardiac cycle, which we demonstrate in Figure 4, 6 and the supplementary videos.
> >
> > Regarding large deformations, this issue was explicitly considered when designing our model. As described in Section 3.4, the vertex update process (Eqs. 3–6) is performed iteratively twice. This is because a single update step is insufficient to accurately handle relatively large deformations, while iterative refinement allows the initial shape to progressively approach the target shape. We provide below and in Appendix F an analysis of the effect of the number of update iterations on performance below on the M\&Ms dataset.
> >
> > |     | 1-slice Myo CD $\downarrow$ | 1-slice LV CD $\downarrow$ | 1-slice RV CD $\downarrow$ |
> > | --- | :---: | :---: | :---: |
> > | Ours (default, repeat=2) | 9.76 | 12.37 | 21.42 |
> > | repeat=1 | 11.36 | 14.31 | 24.26 |
> > | repeat=3 | 9.70 | 12.49 | 21.32 |
> >
> > |     | Full-slice Myo CD $\downarrow$ | Full-slice LV CD $\downarrow$ | Full-slice RV CD $\downarrow$ |
> > | --- | :---: | :---: | :---: |
> > | Ours (default, repeat=2) | 4.11 | 4.85 | 13.68 |
> > | repeat=1 | 4.91 | 5.37 | 15.23 |
> > | repeat=3 | 4.17 | 4.93 | 13.59 |
> >
> > This confirms that using only one update step leads to a noticeable performance drop, whereas increasing the number of iterations beyond two does not provide further improvement. One possible explanation is that cardiac motion itself is physiologically constrained, such that the magnitude of deformation between frames is naturally limited.
> >
> > At the same time, we acknowledge that this design still has limitations. The hyperparameters were chosen given the characteristics of the datasets we currently have. Therefore, when encountering out-of-distribution data, its performance may decline. Meanwhile, if one wants to model other dynamic objects, the appropriate hyperparameters may be different. How to propose a more robust motion model is an interesting study. But for the task at hand, our design already delivers good performance.
> >
> > > **Q4**: Other distillation formulations.
> >
> > **A4**: Thanks for the excellent suggestion. We conducted a preliminary experiment where the motion outputs predicted from the full-slice inputs were directly used to distill the few-slice motion outputs. The corresponding results are shown below.
> >
> > |     | 1-Slice Myo CD $\downarrow$ | 5-Slice Myo CD $\downarrow$ | Full-Slice Myo CD $\downarrow$ |
> > | :---: | :---: | :---: | :---: |
> > | No Distill | 10.34 | 8.12 | 4.03 |
> > | Ours (Feature Distill) | 9.76 | 7.93 | 4.11 |
> > | Motion Distii | 9.86 | 7.90 | 4.08 |
> >
> > This shows that motion-level distillation and feature-level distillation achieve very similar performance, with neither strategy delivering a clear advantage. This observation may suggest that the primary benefit of distillation in our setting comes from transferring structural and motion-related priors from the full-slice observations to the sparse-slice setting, rather than from the specific representation level at which the distillation is applied. In other words, once informative geometric priors are successfully propagated from the dense-input, both intermediate feature supervision and output-level motion supervision appear capable of providing useful guidance under sparse observations.
> >
> > At the same time, we acknowledge that our current experiments on motion-level distillation are still preliminary and may not yet fully explore its potential. Due to time limitations, we were unable to investigate it more thoroughly in the current revision, but we believe they constitute an interesting avenue for future work.
> >
> > Nevertheless, our main intention in introducing distillation was to demonstrate that transferring prior information from complete observations is beneficial when only incomplete sparse inputs are available, and the feature and motion distillation experiments both support this conclusion.

---

### Review · Reviewer_kFsd · 2026-05-14

**Summary Of Contributions:**

Authors present TetHeart, a unified framework for 4D cardiac mesh reconstruction from both full-stack cardiac MRI and sparse real-time slices acquired during interventions. The method combines deformable tetrahedral representations with an attentive feature aggregation module (AFA) that fuses arbitrary sparse 2D observations into a volumetric representation. The framework is trained with a weakly supervised strategy using only ED/ES annotations and includes a distillation mechanism to improve sparse-slice robustness. The paper is evaluated on three public datasets (ACDC, M&Ms, M&Ms-2), as well as external validation on an interventional MRI dataset and the public 4DM dataset without retraining. Results show consistent performance in both full-stack and sparse-slice settings.

**Additional Comments:**

Minor comments:
- Fig. 2 does not clearly visualize where cross-attention is applied, despite cross-attention being central to the method description.
- The volume–time curves in Fig. 4 are difficult to read because they are too small.
- Citation formatting is inconsistent (\cite{} vs \citet{} style usage).
- In Section 3.1, the acronym “SDF” is used before being explicitly defined.
- There are several minor grammatical issues throughout the paper, for example: “performs worser than us”, “proves our model performs best”, “due to it can capture”.

**Audience:**

Yes

**Audience Explanation:**

Authors address a realistic interventional scenario that is currently underexplored. The combination of sparse real-time MRI acquisition with end-to-end 4D reconstruction is technically interesting and clinically relevant. Even beyond cardiac imaging, the proposed ideas for sparse-to-dense reconstruction and adaptive feature aggregation could inspire related work in other dynamic (medical) imaging applications.

**Broader Impact Concerns:**

No broader impact concerns.

**Claims And Evidence:**

Yes

**Claims Explanation:**

The paper provides convincing evidence that the proposed framework can reconstruct 4D cardiac motion from sparse MRI observations, including highly sparse settings such as single-slice inputs. The evaluation is overall strong, with multiple public datasets, external validation, clinically relevant metrics, and qualitative analyses.
However, the quantitative evaluation would benefit from better reporting of statistical significance and uncertainty estimates (e.g., repeated runs, confidence intervals, error bars), especially since some improvements over strong baselines are relatively small.
Overall, the evidence supports the approach's feasibility and usefulness.

**Requested Changes:**

Non-critical requested changes:
- Better report statistical significance and uncertainty estimates throughout the paper.
Most quantitative results are presented as single values without confidence intervals, standard deviations, repeated runs, or cross-validation details. While Appendix D includes paired t-tests, some of this analysis should appear in the main paper. In particular: add uncertainty estimates/error bars in plots such as Fig. 5, report variability for ablations as well, clarify whether results come from reruns, fixed splits, or cross-validation. The problem setting is novel and clinically relevant, so strict statistical superiority is less critical, but the current presentation could appear to overstate the strength of the quantitative gains.
- Better justify the shared weights between the reconstruction and motion branches.
This design choice is interesting but somewhat surprising since static reconstruction and motion estimation are different tasks, and sparse online slices likely differ from dense offline stacks in both distribution and objective. Although the ablation suggests sharing is beneficial, the paper would benefit from stronger intuition, theoretical motivation, or additional analysis/visualization explaining why this sharing works well.
- Strengthen the ablation section.
Current ablations mostly rely on removing individual components and remain relatively limited given the complexity of the framework (AFA, sparse/full distillation, shared encoder, tetrahedral representation, weak supervision, augmentation strategy, etc.). Additional discussion and/or analyses could strengthen the paper, e.g., sensitivity to slice location (not only slice count), impact of distillation in full-stack-only training, runtime/latency tradeoffs under different settings, and qualitative failure cases.

---

> ### Author Response · Authors · 2026-05-28
> **Rebuttal**
>
> We appreciate the reviewer’s thorough evaluation and positive comments on the novelty and clinical relevance of our work. We are encouraged that the reviewer recognized the effectiveness of TetHeart for 4D cardiac mesh reconstruction. We would like to address the concerns below.
>
> > **Q1**: Statistical significance and uncertainty estimates.
>
> **A1**: We have updated Fig. 5 by adding error bars to better reflect the uncertainty and variability of the results. However, since the main paper already contains a large amount of numerical content and the table fonts are already small, we chose to give the statistical significance analysis in appendix instead of the main body of the paper. Thus, we have expanded the corresponding appendix section to include additional analyses, including significance tests for the ablation studies. We have also added pointers to these results in the main paper. We also clarify that all reported results are obtained using 5-fold cross-validation. The quantitative metrics are therefore aggregated across the five folds rather than being derived from a single fixed split or a single training run.
>
> > **Q2**: Shared weights between the reconstruction and motion branches.
>
> **A2**: Thanks for validating the model design and for finding it interesting. Below, we first clarify the weight-sharing ablation experiment in Section 4.5, when we write "when the motion branch is trained without parameter sharing and randomly initialized, ...", the key is the random initialization. This prevents the motion branch from inheriting the rich priors learned in the reconstruction branch. In fact, we can also initialize the motion branch from the parameters of the reconstruction branch without sharing the weights and optimize the two branches separately. However, our experiments show that this approach did not help. Therefore, we adopted the design of shared weight, reducing the number of parameters while having the same performance. We have added the results in Appendix F.
>
> We now provide a more detailed explanation and intuition. Although static reconstruction and motion estimation are conceptually different tasks, they are both concerned with predicting vertex-level geometric properties in a common 3D deformable representation space. In our framework based on deformable tetrahedral meshes, this connection becomes particularly explicit: Static reconstruction predicts vertex offsets and SDF-related attributes on the template tetrahedra $\mathcal{G}^{init}$, while motion estimation predicts vertex displacements of the tetrahedra $\mathcal{G}^{0}$. In both cases, the underlying objective is to infer meaningful vertex-wise geometric transformations conditioned on image observations.
>
> From this perspective, the two branches can be interpreted as learning two stages of the same geometric inference process. Empirically, this shared representation helps transfer priors learned from the reconstruction task to the more challenging motion estimation task. Another important aspect of our design is that parameter sharing is limited to the feature extraction backbone, while the task-specific reasoning modules are explicitly decoupled: The static branch uses a GCN-based decoder, whereas the motion branch employs a GRU-GCN structure. This separation preserves task-specific modeling capacity.
>
> Overall, we find that shared weights provide a simple yet effective way to couple two closely related geometric prediction tasks, improving parameter efficiency without sacrificing expressiveness.
>
> > **Q3**: Strengthening ablation section.
>
> **A3**: Thanks for your suggestion. To strengthen the ablation study, we now provide several additional ablation results in Appendix F. We note that the impact of the AFA configuration on performance and speed was already included in the original manuscript. Beyond this existing analysis, we add a slice position ablation study to investigate the sensitivity of the model to the spatial location of the input slices, complementing the slice-count analysis presented in the main paper. We also include an experiment that removes the distillation loss under the full-stack-only training setting. In addition, ablation studies requested by other reviewers have also been incorporated.
>
> To show the limitations of the technique, we also provide a qualitatively not-so-successful case in the appendix. It features scenarios in which the prediction is less accurate, which should offer insights for future work on particularly challenging cases.
>
> > **Q4**: Minor comments.
>
> **A4**: We have updated Figure 2 to better illustrate what the query, keys, and value in the AFA cross-attention are. The resolution of Figure 4 is also increased for better visibility. We checked that the citations in the main text consistently use the \cite{} format and we have fixed the typos. If there are still some incorrect citations, we would appreciate it if the reviewer could point them out to us.

---

### Decision · Action_Editor_UB4s · 2026-06-27

**Recommendation:** Accept with minor revision

**Additional Comments:**

The authors should make sure they address the Requested Changes made by the reviewers. Also they should clarify the reproducibility issues mentioned above

**Audience:**

Yes

**Audience Explanation:**

The reviews indicate that the paper presents findings that would be of interest to at least a subset of the conference audience, particularly those working in medical imaging, computational cardiology, and clinically oriented machine learning. The main disagreement concerns the degree of methodological novelty rather than the relevance or interest of the findings themselves. It is not clear to me whether source code would be provided to make the results reproducible.

**Claims And Evidence:**

Yes

**Claims Explanation:**

The reviewers broadly agree that the paper addresses an important and clinically relevant problem, is technically sound, and presents convincing empirical results that were further strengthened during rebuttal. However, multiple reviewers also agree that the methodological novelty is limited, making the work more compelling from an application and clinical perspective than as a major algorithmic contribution for a top-tier machine learning conference. As a result it is better aligned with application-oriented venues or journals than top-tier ML conferences.
It is not clear to me whether source code would be provided to make the results reproducible. It is also not clear whether the novel dataset mentioned will be provided online. Doing so would significantly increase the paper's impact.